# Llamas Know What GPTs Don't Show: Surrogate Models for Selective Classification

## Abstract

To maintain user trust, large language models (LLMs) should signal low confidence on examples they get incorrect, instead of misleading the user. The standard approach of estimating confidence is to use the softmax probabilities of these models, but state-of-the-art LLMs such as GPT-4 and Claude do not provide access to these probabilities. We first study eliciting confidence linguistically—asking an LLM for its confidence in its answer—but we find that this leaves a lot of room for improvement (79% AUC on GPT-4 averaged across 12 question-answering datasets—only 5% above a random baseline). We then explore using a *surrogate* confidence model—using a model where we do have probabilities to evaluate the original model's confidence in a given question. Surprisingly, even though these probabilities come from a different model, this method leads to higher AUC than linguistic confidences on 10 out of 12 datasets. Our best method mixing linguistic confidences and surrogate model probabilities gives state-of-the-art performance on all 12 datasets (85% average AUC on GPT-4).

## 1 Introduction

As large language models (LLMs) are increasingly deployed, it is important that they signal low confidence on examples where they make mistakes. This problem is called selective classification (or classification with a reject option) and is widely studied in machine learning (Cordella et al., 1995; Geifman & El-Yaniv, 2017; Feng et al., 2019; Jones et al., 2021), learning theory (El-Yaniv & Wiener, 2010; Bartlett & Wegkamp, 2008), and natural language processing (Kamath et al., 2020; Liang et al., 2022; Xiong et al., 2023). Traditional approaches leverage the model's softmax probability (Hendrycks & Gimpel, 2017; Jones et al., 2021; Liang et al., 2022) or the model's representations (Lee et al., 2018). This paper's goal is to produce *good confidence estimates for state-of-the-art LLMs* such as GPT-4 and Claude, which *do not provide model probabilities* or representations.

We first examine a natural idea of eliciting linguistic confidence scores (Tian et al., 2023; Lin et al., 2022; Xiong et al., 2023)—prompting the LLM to ask for its confidence that its answer is correct (Figure 1, GPT-4 Linguistic). We find that while linguistic confidences work better than a random guessing baseline, it leaves a lot of room for improvement. Our results hold across 12 standard datasets (8 MMLU datasets, TruthfulQA, CommonsenseQA, OpenbookQA, and MedQA), 5 models (GPT-4, Claude, GPT-3.5, Llama-2, and text-davinci-003), and 24 different prompt formats (e.g., chain-of-thought, different instructions, fake few-shot prompts). Averaged across the datasets, GPT-4 achieves a modest selective classification AUC of 78.8%, which is only 5.3% above a random guessing baseline. When model probabilities are available (for less accurate models such as Llama-2), the linguistic confidences perform much worse than using model probabilities (60.9% vs. 73.1% AUC on Llama-2).

Instead, we propose a surrogate model approach of taking the answer from GPT-4 or Claude, but the *confidence from a different model* such as Llama 2 (Figure 1, Surrogate). The surrogate model method improves the average selective classification AUC for GPT-4 to 81.8%. Even using a much smaller Llama-13B model improves the AUC for models such as GPT-4, Claude, and GPT-3.5. Intriguingly, the model generating the confidence score is different (and much worse) than the model generating the answer, but its confidence scores transfer over.

We find that the linguistic confidence scores are still useful: adding these scores to the surrogate model's probabilities leads to further gains (Figure 1, Mixture). For example, this mixture method increases the selective classification AUC of GPT-4 to 83.0%. The mixture method also outperforms

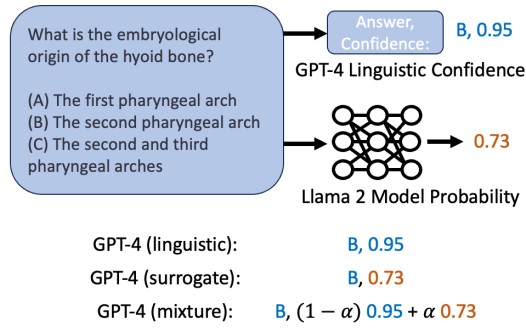

| | Selective Classification AUC | Accuracy |
|---|---|---|
| Llama-2 (linguistic) | 0.61 | 0.58 |
| Llama-2 (model prob) | 0.73 | 0.58 |
| GPT-4 (linguistic) | 0.79 | 0.74 |
| GPT-4 (surrogate) | 0.82 | 0.74 |
| GPT-4 (mixture) | 0.83 | 0.74 |
| GPT-4 (mixture++) | **0.85** | 0.74 |

(a) Proposed methods for eliciting confidence scores

(b) Performance of methods averaged over 12 datasets

Figure 1: Our goal is to provide good confidence estimates for state-of-the-art LLMs such as GPT-4 and Claude which do not give access to internal model probabilities. One natural approach (GPT-4 linguistic) is to prompt the model and ask for its confidence. Interestingly, we find that taking the answer from GPT-4, but the internal model probability from a different surrogate model (e.g., an open model such as Llama-2) gets even better results (0.82 AUC). Mixing GPT-4's linguistic probabilities with the surrogate model probabilities gives further gains (0.83 AUC). Our AUC numbers are better than concurrent work (Xiong et al., 2023), but combining these approaches leads to the best results (Mixture++; 0.85 AUC). Our findings also hold for Claude and GPT-3.5 (Section 4 and 5).

concurrent work (Xiong et al., 2023) on self-consistency (AUC: 82.6%), which is more expensive (involves sampling GPT-4 five times per input) and involves post-processing. Combining our method with self-consistency leads to the *best results: average AUC of 84.5%*.

Our analysis suggests that linguistic confidence scores do not work well by themselves because they are very coarse-grained—for example, GPT-4 outputs the exact same confidence (0.9) on 50% of examples, which limits its ability to separate correct and incorrect answers. Surrogate model probabilities appear to work well even on a different model, because the examples that are challenging for one model transfer over to a different model. Finally, mixing in just a small fraction of surrogate model probabilities allow answers which previously had the same linguistic confidence to be separable through different composite confidence scores, boosting the overall performance.

## 2 SETUP

Our goal is selective classification: outputting confidence scores that are higher on inputs where the model is correct, than inputs where the model is incorrect (El-Yaniv & Wiener, 2010; Geifman & El-Yaniv, 2017). We focus on state-of-the-art language models such as GPT-4 and Claude, which do not expose probabilities computed in their softmax output layer.

**Task.** Given a text input $x$, a model outputs a (possibly stochastic) answer $y(x)$. Let $R(x, y) = 1$ if an answer $y$ is correct for input $x$, and 0 otherwise. Our goal is to output a *confidence score* $C(x) \in [0, 1]$. Good confidence scores are essential in real world machine learning systems: for inputs when $C(x)$ is lower, we can defer to a human expert or alert the user, instead of misleading the user with an incorrect answer.

**Metrics.** A popular metric for selective classification is the *AUC* (area under the coverage-accuracy curve) (El-Yaniv & Wiener, 2010; Liang et al., 2022), which examines how accurate the model is if allowed to abstain (say "I don't know") on some examples. Let $A(c)$ be the selective accuracy at coverage $c$: the accuracy if the model only makes a prediction on the $c$ proportion of data with highest confidence scores[1]. The AUC is the average selective accuracy $A(c)$ over all $c$:

$$\text{AUC}(C, y) = \int_0^1 A(c) dc \tag{2.1}$$

A random baseline (outputting uniform random probabilities for each input) achieves $\text{AUC}(C, y) =$ Accuracy, so a model with good confidence scores should achieve a higher AUC than accuracy.

---

[1]The classifier used to select the $c$ proportion of examples can be deterministic or randomized. We report results with a deterministic classifier, but also compute results with a randomized classifier in A.16.

We also examine the *AUROC (area under the receiver operator curve)*, a standard metric (Hendrycks & Gimpel, 2017; Xiong et al., 2023) which examines how well the confidence scores can distinguish between correct and incorrect examples. Outputting random confidence scores gets an AUROC of 0.5, so a model with good confidence scores should achieve AUROC above 0.5.

We also report *ECE (expected calibration error)* numbers in Appendix A.6. ECE examines if a model's confidence aligns with its accuracy, but does not indicate the model's ability to distinguish between correct and incorrect examples, so we focus on the AUC and AUROC metrics.[2]

**Datasets.** We study model performance and confidence on twelve standard question answering datasets: *TruthfulQA* (TQA) (Lin et al., 2021), *CommonsenseQA* (CSQA) (Talmor et al., 2019), *OpenbookQA* (OBQA) (Mihaylov et al., 2018), *MedQA* (Jin et al., 2021), and 8 *MMLU* (Hendrycks et al., 2021) datasets - professional law (Law), business ethics (Ethics), conceptual physics (Physics), econometrics (Econ), abstract algebra (Algebra), college chemistry (Chem), computer security (Security), and US Foreign Policy (Policy). These datasets span several diverse categories including math reasoning, scientific knowledge, computer science, social science, and commonsense reasoning. We sample 250 questions from the test split of each dataset to report results on (if the test set is smaller, we use the full test set). See Appendix A.1 for more details.

**Models.** We study state-of-the-art language models, most of which *do not* provide access to internal probabilities as of the writing of this paper - *GPT-4* (OpenAI, 2023a), *Claude-v1.3* (Anthropic, 2023), and *GPT-3.5-Turbo* (OpenAI, 2022) (June 13th, 2023, snapshot). We also study a few recent models which *do* provide model probabilities for systematic comparisons—*Llama-2* and *Llama-2-chat* (70b and 13b sizes) (Touvron et al., 2023) and *text-davinci-003* OpenAI (2023b). If Llama-2 is mentioned in the text without further identifiers, we refer to the Llama-2-70b base model.

### 2.1 CONFIDENCE ELICITATION METHODS

**Linguistic Confidences.** We zero-shot prompt models to elicit linguistic confidences, sampling the answer and confidence together in a single generation, greedily at temperature $T = 0$. We experiment with 24 different prompts across various categories (chain-of-thought, different instructions, fake few shot examples) and found the results to be consistent across prompts. We report results on our best prompt (see more details on the exact prompt in Appendix A.3).

**Model Probabilities.** Weaker models such as Llama-2 and text-davinci-003 provide token-level probabilities for text. We let the confidence score be the probability of the generated answer choice.

**Surrogate models for confidences.** Since models such as GPT-4 do not give a confidence estimate, we propose using a surrogate model (e.g., Llama-2) to provide confidence estimates. Formally, given an input $x$ we output $y(x) = y_{\mathsf{gpt\text{-}4}}(x)$ (GPT-4's answer) and $C(x) = C_{\mathsf{llama\text{-}2}}(x)$ (Llama-2's confidence). Even though these confidence scores come from a *different* model, Section 4 shows that the surrogate method outperforms linguistic confidence scores.

**Mixture of models.** We also propose a mixture of models method where we combine the linguistic confidence from the main model and the surrogate model's confidence score: given input $x$ we output $(1-\alpha)C_M(x) + \alpha C_S(x)$ where $M$ is the main model and $S$ is the surrogate model (Algorithm 1). We use Llama-2-70b as our surrogate model since that performs the best, and optimize $\alpha$ to minimize AUC. Interestingly, in Section 5, we show that even $\alpha = 0.001$ works well. For details on the optimal $\alpha$ values for each model, see Appendix A.8.

---

**Algorithm 1: Mixture of Models Confidence**

**Data:** A question $x$
**Result:** A prediction $\widehat{y}$, a confidence score $c$
$\widehat{y}, c_1 = \mathtt{MainModel}(x)$;
$c_2 = \mathtt{SurrogateModel}(x)$;
$c = (1 - \alpha)c_1 + \alpha c_2$;

---

## 3 LINGUISTIC CONFIDENCES: ASKING THE MODEL FOR ITS CONFIDENCE

State-of-the-art language models such as GPT-4 and Claude do not give access to internal model probabilities. In this section, we examine linguistically eliciting confidence: prompt models to assign their answers a confidence score between 0 and 1. We find that these linguistic confidences

---

[2]Intuitively, calibration requires that if we output a 0.6 confidence on 100 examples, then we should get $0.6 \cdot 100 = 60$ of them correct. For a classifier with accuracy $A$, one (degenerate) way to have perfect calibration (best possible ECE) is to output confidence $C(x) = A$ for every example $x$.

| Metric | Confidence Type | TQA | MedQA | CSQA | OBQA | Law | Ethics | Physics |
|---|---|---|---|---|---|---|---|---|
| **AUC** | Text-davinci Linguistic | 0.484 | 0.504 | 0.713 | 0.775 | 0.532 | 0.590 | 0.579 |
| | Text-davinci Prob | **0.608** | **0.656** | **0.861** | **0.929** | **0.714** | **0.783** | **0.697** |
| | Llama-2 Linguistic | 0.583 | 0.584 | 0.689 | 0.803 | 0.581 | 0.639 | 0.628 |
| | Llama-2 Prob | **0.711** | **0.735** | **0.804** | **0.923** | **0.749** | **0.834** | **0.763** |
| | GPT-3.5 Linguistic | 0.590 | 0.536 | 0.684 | 0.781 | 0.508 | 0.674 | 0.526 |
| | Claude Linguistic | 0.679 | 0.687 | **0.788** | 0.874 | 0.647 | **0.863** | 0.713 |
| | GPT-4 Linguistic | **0.883** | **0.828** | 0.786 | **0.958** | **0.711** | 0.850 | **0.819** |
| **AUROC** | Text-davinci Linguistic | 0.525 | 0.500 | 0.503 | 0.509 | 0.500 | 0.500 | 0.500 |
| | Text-davinci Prob | **0.718** | **0.696** | **0.806** | **0.840** | **0.715** | **0.758** | **0.637** |
| | Llama-2 Linguistic | 0.618 | 0.541 | 0.555 | 0.484 | 0.517 | 0.602 | 0.593 |
| | Llama-2 Prob | **0.745** | **0.722** | **0.731** | **0.777** | **0.733** | **0.868** | **0.732** |
| | GPT-3.5 Linguistic | 0.535 | 0.500 | 0.520 | 0.516 | 0.508 | 0.509 | 0.504 |
| | Claude Linguistic | **0.669** | 0.584 | **0.631** | 0.647 | 0.586 | **0.760** | **0.652** |
| | GPT-4 Linguistic | 0.665 | **0.716** | 0.547 | **0.656** | **0.591** | 0.720 | 0.522 |

| Metric | Confidence Type | Econ | Algebra | Chem | Security | Policy | **Avg** |
|---|---|---|---|---|---|---|---|
| **AUC** | Text-davinci Linguistic | 0.412 | 0.300 | 0.440 | 0.690 | 0.851 | 0.573 |
| | Text-davinci Prob | **0.431** | **0.338** | **0.644** | **0.890** | **0.939** | **0.708** |
| | Llama-2 Linguistic | 0.402 | 0.212 | 0.470 | 0.792 | 0.922 | 0.609 |
| | Llama-2 Prob | **0.498** | **0.263** | **0.647** | **0.866** | **0.981** | **0.731** |
| | GPT-3.5 Linguistic | 0.412 | 0.330 | 0.460 | 0.737 | 0.806 | 0.587 |
| | Claude Linguistic | 0.621 | 0.339 | 0.591 | 0.823 | 0.918 | 0.712 |
| | GPT-4 Linguistic | **0.628** | **0.534** | **0.616** | **0.886** | **0.955** | **0.788** |
| **AUROC** | Text-davinci Linguistic | 0.500 | 0.500 | 0.500 | 0.500 | 0.506 | 0.504 |
| | Text-davinci Prob | **0.549** | **0.532** | **0.695** | **0.858** | **0.795** | **0.717** |
| | Llama-2 Linguistic | 0.533 | 0.424 | 0.520 | 0.613 | 0.576 | 0.548 |
| | Llama-2 Prob | **0.622** | **0.546** | **0.732** | **0.775** | **0.871** | **0.738** |
| | GPT-3.5 Linguistic | 0.518 | 0.522 | 0.500 | 0.519 | 0.519 | 0.514 |
| | Claude Linguistic | **0.573** | 0.543 | 0.668 | 0.687 | 0.645 | 0.637 |
| | GPT-4 Linguistic | 0.551 | **0.599** | **0.691** | **0.750** | **0.753** | **0.647** |

Table 1: **AUC and AUROC - Linguistic Confidences vs Model Probabilities** We compare the AUC and AUROC values for linguistic confidences and model probabilities in weaker models (text-davinci-003 and Llama-2-70b), and find that model probabilities consistently outperform linguistic confidences. For closed source models (which don't provide model probabilities), we see that Claude and GPT-4 provide the best linguistic confidences in both AUC and AUROC. The same model is used to provide both the answer and the confidence estimate.

leave a lot of room for improvement (around 50-65% AUROC, compared to 50% for a random guessing baseline). These linguistic confidences are also much worse than internal model probabilities when available (for weaker models such as text-davinci-003 and Llama-2). We show AUC and AUROC results on all datasets and models in Table 1.

**Linguistic confidences are mediocre.** The AUROC values of linguistic confidences from text-davinci, Llama-2-70b, and GPT-3.5 are close to 50% (Table 1), which is the score achieved by guessing a random confidence, indicating that linguistic confidences are not a reliable means of separating correct and incorrect examples. The linguistic confidences of the strongest models, Claude and GPT-4, are better and result in AUROCs in the 60-65% range, but still leave a lot of room for improvement. AUC values show a similar story: the AUC of linguistic confidences are close to their accuracy (which is the score achieved by a random guessing baseline). Averaged over the datasets,

Llama-2 has an accuracy of 58.8% and AUC of 62.8%; GPT-4 has an accuracy of 73.5% and AUC of 78.8%; and Claude has an accuracy of 65.5% and AUC of 71.3%.

**Linguistic confidences are worse than model probabilities.** The best current models (GPT-4 and Claude) do not provide model probabilities, but we compare the quality of model probabilities and linguistic confidences for text-davinci-003 and the Llama-2 models. For these models, the model probabilities result in better AUC and AUROC values for all of our datasets (Table 1). For Llama-2, the model probabilities achieve a *13.5% higher AUC and 19% higher AUROC* than the linguistic confidences. The Chat model (Llama-2-70b-Chat) shows similar trends (Appendix A.5). We measure the correlation between linguistic confidences and model probabilities in Appendix A.14.

**Linguistic confidences are robust to prompts.** We examine linguistic confidences using 24 distinct prompts, including asking for numerical confidence or probability scores, asking the model to categorize its confidence into 'not sure', 'sure', and 'very sure', allowing the model to explain confidences with chain-of-thought, asking the model for its confidence in a follow-up question, and varying the prompt instructions. We show results for the best prompt, as there was very little difference in performance across prompts—our results hold for other prompts as well (Appendix A.3).

**Linguistic confidences improve with scale, but not enough.** The quality of linguistic confidences improves with model scale. We see that GPT-4 and Claude have the best linguistic confidences, followed by the Llama-2-70b models, GPT-3.5, and finally text-davinci-003. While the *linguistic confidences* from GPT-4 are not bad (65% average AUROC), they are worse than *model probabilities* from Llama-2-70b (74%) and even text-davinci-003 (72%). Note that AUC scores increase with accuracy—GPT-4 Linguistic has the highest AUC because GPT-4 has much higher accuracy than Llama-2. The overall utility of a selective classifier depends on both its accuracy and confidence quality, so in the next section we examine ways to improve the confidences of GPT-4 and Claude.

## 4 SURROGATE MODELS ARE RELIABLE CONFIDENCE ESTIMATORS

In the previous section we found that linguistic confidences leave a lot of room for improvement. Here we show that model probabilities from a separate 'surrogate' model can surprisingly provide better confidence estimates for a model than its own linguistic confidence scores, even though the probabilities come from a different model.

### 4.1 RESULTS

**Surrogate model confidences outperform linguistic confidences.** AUC improves for all models when probabilities from any surrogate model are used, as opposed to using the model's own linguistic confidences. Figure 8 shows a heatmap of the AUC for different main models (on the $x$-axis) as we vary the surrogate model (on the $y$-axis). We see that model probabilities (bottom three rows) lead to higher AUC (are darker) than linguistic confidences (top 5 rows) even when the probabilities come from a different model. For example, using Llama-2-70B probabilities as a surrogate improves AUC from 78.8% to 81.8% for GPT-4, 71.2% to 75.6% for Claude, and 58.7% to 71.9% for GPT-3.5, and the AUROC also shows similar increases for all models (Table 2, Figure 3).

**Weak surrogates are also good confidence estimators.** Even using Llama-2-13B or text-davinci-003 as a surrogate leads to comparable or better performance than using a model's own linguistic confidences. We found this intriguing because these models are much smaller and less accurate, e.g., Llama-2-13B has an average accuracy of 47.2% vs. 65.5% for Claude and 73.5% for GPT-4. As we might expect, better models (such as Llama-2-70b) are better surrogates.

**Other findings.** Recent work suggests RLHF'ed chat models may be less calibrated than base models. Llama-2-70B Base has better linguistic confidences and probabilities than the Chat model, but both perform similarly as surrogates (Appendix A.11, A.5). Finally, we find that *linguistic confidences* from stronger models can be good surrogates for weaker models—the AUC of text-davinci-003 improves by 5% when using GPT-4's linguistic confidences instead of its own.

## 5 MIXTURES OF MODELS FOR BETTER CONFIDENCE ESTIMATES

In the previous section, we proposed the use of surrogate models—using a main model to produce answers and a separate, surrogate to estimate the main model's confidence in the answers—and found surrogates to outperform linguistic confidence scores elicited from the main model. In this

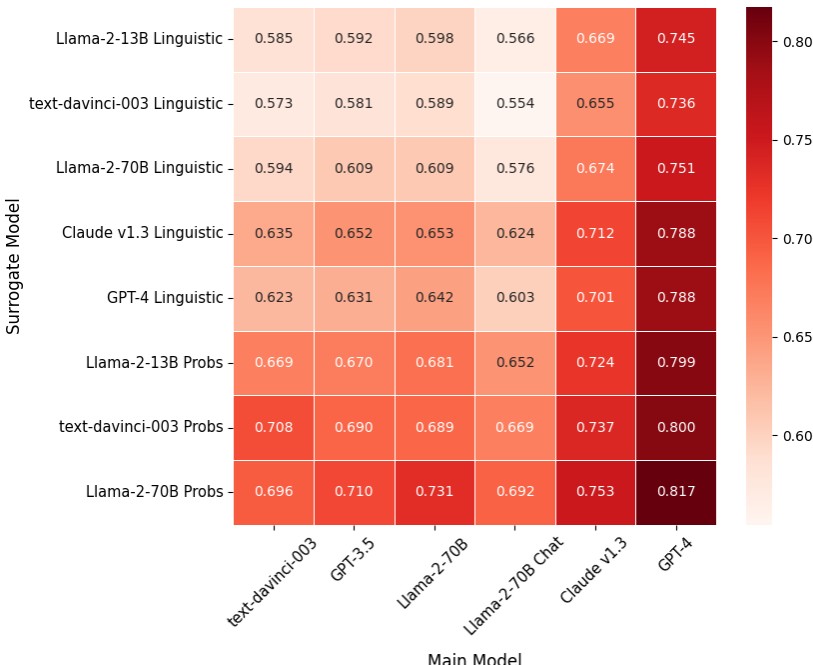

Figure 2: **AUCs for Different Surrogate Models.** We plot the AUC as we vary the main model (on the $x$-axis) and the surrogate model (on the $y$-axis). Using surrogate model probabilities as confidence estimates improves AUCs for all models over their own linguistic confidences—the bottom 3 rows (surrogate probabilities) are darker than the top 5 rows (linguistic confidences). Even model probabilities from a smaller and less accurate Llama-13B model improves AUCs for all models.

|  |  | Text-davinci | GPT-3.5 | Llama-2 | Claude | GPT-4 |
|---|---|---|---|---|---|---|
| **AUC** | Ling. Conf. | 0.573 | 0.587 | 0.609 | 0.712 | 0.788 |
|  | Surrogate[†] | 0.708 | 0.719 | **0.731** | 0.753 | 0.818 |
|  | Tiebreak[†] | **0.711** | 0.718 | 0.715 | 0.754 | 0.826 |
|  | Mixture of Models[†] | **0.711** | **0.721** | **0.731** | **0.762** | **0.830** |
| **AUROC** | Ling. Conf. | 0.504 | 0.515 | 0.548 | 0.637 | 0.647 |
|  | Surrogate[†] | 0.717 | 0.705 | **0.738** | 0.671 | 0.654 |
|  | Tiebreak[†] | **0.719** | 0.707 | 0.700 | 0.677 | 0.689 |
|  | Mixture of Models[†] | **0.719** | **0.708** | 0.737 | **0.685** | **0.696** |

Table 2: **AUC and AUROC of Surrogate and Mixture of Model Methods.** We compare the performance of our proposed methods[†] with the baseline linguistic confidence method (gray). For both AUC and AUROC, our proposed methods outperform linguistic confidences on all models. Mixture of models improves the AUC of Claude by 5% and GPT-4 by 4.2%.

section, we find that the linguistic confidences are still useful—they can be composed with a surrogate model's confidences to get state of the art confidence estimates for all models.

## 5.1 RESULTS

**Mixtures of models provide best confidences.** Mixing surrogate and linguistic confidences (Algorithm 1) leads to the best confidence estimates for all models—AUCs increase from 78.7% to 83.0% for GPT-4 and 71.2% to 76.2% for Claude. AUROCs also increase for these models, by 5.2% for GPT-4 and 4.8% for Claude (Table 2). We plot selective accuracy against coverage, where the mixture and surrogate method lie above the linguistic confidences curve (Figure 3).

| | Method | TQA | MedQA | CSQA | OBQA | Law | Ethics | Physics |
|---|---|---|---|---|---|---|---|---|
| | Ling. Conf. | 0.883 | 0.828 | 0.786 | 0.958 | 0.711 | 0.850 | 0.819 |
| | SC Vanilla Ling. Conf. | 0.857 | 0.763 | 0.798 | 0.942 | 0.693 | 0.785 | 0.830 |
| | SC Hybrid Ling. Conf. | 0.902 | **0.883** | 0.837 | 0.977 | 0.729 | **0.900** | 0.845 |
| AUC | Surrogate[†] | 0.866 | 0.844 | 0.843 | 0.965 | 0.762 | 0.848 | **0.891** |
| | Tiebreak[†] | 0.902 | 0.871 | 0.826 | 0.967 | 0.768 | 0.889 | 0.861 |
| | Mixture[†] | 0.895 | 0.864 | 0.842 | 0.969 | **0.780** | 0.881 | 0.886 |
| | SC Mixture[†] | **0.921** | 0.873 | **0.873** | **0.979** | 0.757 | 0.894 | 0.881 |
| | Ling. Conf. | 0.665 | 0.716 | 0.547 | 0.656 | 0.591 | 0.720 | 0.522 |
| | SC Vanilla Ling. Conf. | 0.549 | 0.550 | 0.540 | 0.607 | 0.571 | 0.612 | 0.570 |
| | SC Hybrid Ling. Conf. | 0.698 | **0.767** | 0.623 | 0.833 | 0.619 | **0.817** | 0.592 |
| AUROC | Surrogate[†] | 0.543 | 0.666 | 0.652 | 0.683 | 0.619 | 0.617 | 0.648 |
| | Tiebreak[†] | 0.671 | 0.750 | 0.607 | 0.716 | 0.628 | 0.740 | 0.589 |
| | Mixture[†] | 0.642 | 0.731 | 0.642 | 0.731 | 0.655 | 0.711 | 0.648 |
| | SC Mixture[†] | **0.702** | 0.747 | **0.677** | **0.838** | **0.655** | 0.783 | **0.663** |

| | Method | Econ | Algebra | Chem | Security | Policy | **Avg** |
|---|---|---|---|---|---|---|---|
| | Ling. Conf. | 0.628 | 0.534 | 0.616 | 0.886 | 0.955 | 0.788 |
| | SC Vanilla Ling. Conf. | 0.619 | 0.575 | 0.639 | 0.860 | 0.919 | 0.773 |
| | SC Hybrid Ling. Conf. | 0.658 | 0.585 | 0.726 | 0.912 | 0.964 | 0.826 |
| AUC | Surrogate[†] | **0.667** | 0.571 | 0.693 | 0.888 | 0.971 | 0.817 |
| | Tiebreak[†] | 0.654 | 0.581 | 0.714 | 0.910 | 0.974 | 0.826 |
| | Mixture[†] | 0.664 | 0.581 | 0.718 | 0.908 | **0.976** | 0.830 |
| | SC Mixture[†] | 0.657 | **0.645** | **0.763** | **0.926** | 0.973 | **0.845** |
| | Ling. Conf. | 0.551 | 0.599 | 0.691 | 0.750 | 0.753 | 0.647 |
| | SC Vanilla Ling. Conf. | 0.562 | 0.662 | 0.699 | 0.634 | 0.556 | 0.593 |
| | SC Hybrid Ling. Conf. | **0.617** | 0.682 | 0.818 | 0.798 | 0.755 | 0.718 |
| AUROC | Surrogate[†] | 0.578 | 0.621 | 0.676 | 0.779 | 0.764 | 0.654 |
| | Tiebreak[†] | 0.569 | 0.648 | 0.730 | 0.815 | 0.805 | 0.689 |
| | Mixture[†] | 0.578 | 0.648 | 0.729 | 0.814 | **0.822** | 0.696 |
| | SC Mixture[†] | 0.590 | **0.763** | **0.819** | **0.839** | 0.810 | **0.740** |

Table 3: **AUC and AUROC of All Confidence Methods for GPT-4.** Our proposed surrogate model method outperforms linguistic confidences on 10/12 datasets on AUC. Mixing surrogate probabilities and linguistic confidences outperforms vanilla linguistic confidences on all 12 datasets. The mixture of surrogate probabilities also outperforms self-consistency and hybrid self-consistency confidences, the best method in Xiong et al. (2023), on average (AUC 83.0% vs. 82.6%). Mixing surrogate probabilities with hybrid self-consistency linguistic confidences leads to the best confidence estimates overall, outperforming all methods with an average 84.5% AUC and 74.0% AUROC, which is a gain of 5.7% and 9.3% respectively over vanilla linguistic confidences.

**Epsilon is all you need.** We study a special case of Algorithm 1 called Tiebreaking, where we set $\alpha$ to a small value $\epsilon \to 0$. Adding only 0.1% of a surrogate model's probabilities to a model's linguistic confidences performs better than linguistic confidences or surrogate probabilities alone, and closely matches performance of the optimal $\alpha$ (Table 2). For GPT-4, Tiebreaking achieves 90% of the AUC gains (over linguistic confidences) of the optimal $\alpha$, and 85.7% of the AUROC gains.

**Mixing surrogate and self-consistency confidences leads to further gains.** Concurrent work (Xiong et al., 2023) on eliciting linguistic confidences uses self-consistency (SC) to sample multiple linguistic confidence scores for each answer and aggregates them through a post processing technique (SC Hybrid). This method is more expensive than our surrogate method and only works with additional post-processing steps (Appendix A.7). We experiment with leveraging these SC-based linguistic confidences for GPT-4 in $c_1$ in Algorithm 1. This leads to state-of-the-art confidence estimates, also outperforming their hybrid self-consistency technique (Table 3), with an 5.7% gain in AUC for GPT-4 over vanilla linguistic confidences, and a 9.3% gain in AUROC.

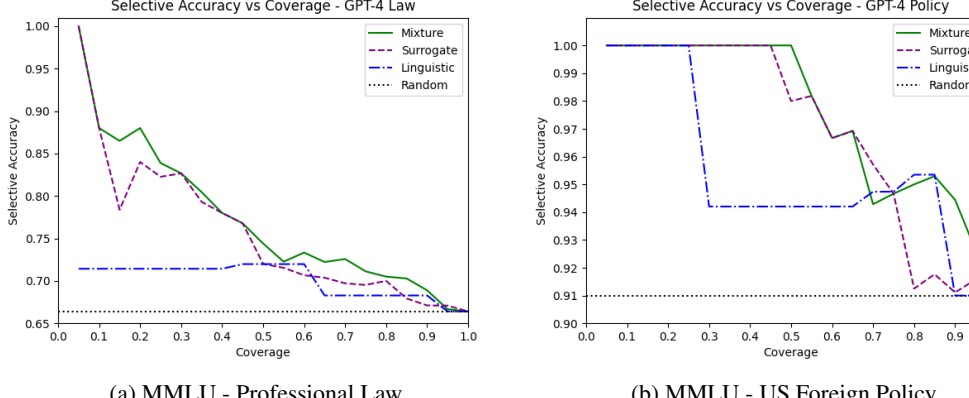

Figure 3: **Selective Accuracy vs. Coverage for GPT-4.** Our surrogate and mixture methods have a higher area under the selective accuracy vs coverage curve (AUC) than the linguistic confidence and random confidence baselines. We plot the coverage $c$ on the $x$-axis and the selective accuracy (accuracy on the top $c$ fraction of examples) on the $y$-axis, for two representative tasks. Notice that the mixture (green solid) and surrogate (purple dashed) lines are above the linguistic confidence (blue dashed/dotted) and random guessing baseline (black dotted).

**Other findings.** Probabilities of smaller surrogate models can also be composed with linguistic confidences—composing Llama-2-13b's probabilities with GPT-4's linguistic confidences retains 88% of the AUC gains seen from composing Llama-2-70b's probabilities. Composing GPT-4 and Claude's linguistic confidences can boost GPT-4's AUC and AUROC by 3% each, indicating that linguistic confidences from different models can also be complementary. Appendix A.10 details experiments on composing confidences from multiple surrogate models.

## 6 ANALYSIS

**Why Are Vanilla Linguistic Confidences Worse Than Model Probabilities?** In Section 3, we showed that linguistic confidences underperformed model probabilities. Here we provide some intuitions for this behavior. We observe that the distribution of model probabilities is quite varied (1456 unique values for Llama-2-70b across 12 datasets), while the distribution of linguistic confidences is clustered (only 8 unique values for GPT-4 across 12 datasets). This clustering may be because training corpora contain high frequencies of "nice" probabilities such as 90% or 100% (Zhou et al., 2023). The repetitiveness of linguistic confidences, compared to model probabilities, hinders relative confidence ordering and good AUC/AUROC performance—GPT-4 repetitively generates 0.9 for 50% of examples across 12 tasks, so it cannot separate them.

**Why Do Surrogate Methods Improve Confidence Scores?** In section 4, we demonstrate that models can receive good quality confidence estimates from other surrogate models. Here we provide some intuitions for our results. We find that for a main model $M$, a model $S$ tends to be a better surrogate when there is a higher correlation in the questions answered correctly by $M$ and $S$. The questions GPT-4 answers correctly are more correlated with those that Llama-2-70b answers correctly (Pearson correlation of 0.39), than those that Llama-2-13b answers correctly (correlation 0.19) (Appendix A.12). In Figure 4, we plot the embeddings of questions that GPT-4 gets incorrect (blue dots) and the questions two potential surrogates Llama-2-70b and Llama-2-13b get incorrect (green dots) (see Appendix A.9 for more details). GPT-4 and Llama-2-70b tend to make mistakes on more of the same questions (more black dots on the left plot). We also see more spatial similarity in their mistakes. So better surrogates $S$ and their corresponding main models $M$ may struggle with *semantically related* concepts, causing them to have low confidences on similar types of questions.

**Why Does Tiebreaking Work Well?** Linguistic confidences tend to be clustered at only a few values (e.g., 0.9), limiting their ability to separate correct and incorrect answers. Since a surrogate model's probabilities for each example are nearly unique, composing just a small fraction of them can allow answers which previously had the same linguistic confidence to now be separable.

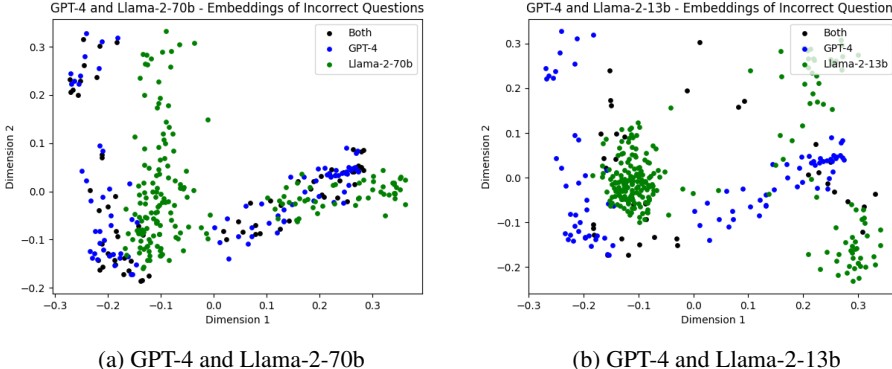

(a) GPT-4 and Llama-2-70b    (b) GPT-4 and Llama-2-13b

Figure 4: **Embeddings of Incorrect Questions for GPT-4 and Surrogate Models** Plots of the embeddings of questions GPT-4 and two surrogate models (Llama-2-70b and Llama-2-13b) answer incorrectly on two representative datasets - TruthfulQA and College Chemistry. Questions only GPT-4 answers incorrectly are in blue, questions GPT-4 and the surrogate answer incorrectly are in black, and questions only the surrogate answers incorrectly are in green. There are more questions that both GPT-4 and Llama-2-70b answer incorrectly and more semantic similarity in their incorrect questions. This indicates that Llama-2-70b and GPT-4 struggle with semantically related concepts and that the 70b model may more closely represent GPT-4's uncertainty than the 13b model.

## 7   RELATED WORK

**Confidence Estimation for LLMs.** Several studies explore confidence estimation for LLMs. Kadavath et al. (2022) show that Claude's probabilities are calibrated on MC tasks, while Zhou et al. (2023) investigate the impact of uncertainty expressions on model accuracy. (Lin et al., 2022) fine-tune LLMs to improve confidence estimates (Appendix A.4). Our work focuses on confidence estimation for models that don't provide log probabilities. Concurrent work (Xiong et al., 2023) studies LLM-generated confidences, but focuses on self-consistency based methods which are expensive in prompting GPT-4 multiple times, and add complexity by requiring post-processing steps. Our methods show gains over their best method and are less expensive, since probabilities from smaller models (Llama-2) are used to improve the confidences of larger models (GPT-4) (Appendix A.7).

**Selective Classification and OOD Detection.** Our paper focuses on selective classification, which is a classical problem in machine learning (El-Yaniv & Wiener, 2010; Khani et al., 2016; Feng et al., 2019; Jones et al., 2021) and statistics (Chow, 1970; Hellman & Raviv, 1970). A related problem is out-of-distribution detection (Pimentel et al., 2014; Liang et al., 2018; Ovadia et al., 2019), where the goal is to detect examples very different from training (where the model may make mistakes). Prior work uses internals of the models—probability outputs (Hendrycks & Gimpel, 2017), model representations (Lee et al., 2018), or even updates the training procedure (Bartlett & Wegkamp, 2008; Mozannar & Sontag, 2020)—which state-of-the-art LLMs do not provide access to.

**Calibration.** The general idea of confidence estimation is also studied in calibration (Murphy & Winkler, 1977; DeGroot & Fienberg, 1983; Naeini et al., 2014; Guo et al., 2017). While related, the focus is different—a model which outputs its accuracy on every example has 0 calibration error (ECE), but cannot *separate* correct and incorrect examples (Kuleshov & Liang, 2015).

**LLM Self-Evaluation.** LLMs have shown progress in improving generations through self-critiques (Madaan et al., 2023; Chang et al., 2023), or validating generations using tools (Gou et al., 2023). Better confidence estimation may allow models to produce more reliable critiques (Appendix A.15).

## 8   CONCLUSION

Our work aims to address the open challenge of eliciting good confidence estimates from state-of-the-art LLMs such as GPT-4 and Claude, which don't provide access to their internal probabilities. We show that using probabilities from weaker surrogate models provide effective confidence estimates for larger models, and find that interpolating these with linguistic confidences leads to even better confidence estimates.

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

# A APPENDIX

## A.1 DATASET DETAILS

**TruthfulQA** is a multiple choice benchmark designed to check the truthfulness of large language models by testing them on questions across 38 different categories like health and politics where humans might provide incorrect responses due to implicit biases or incorrect beliefs. This task is challenging for language models because they may imbibe these same misconceptions from training corpora.

**MedQA** is a challenging dataset testing medical knowledge with questions based on the United States Medical License Exams (USMLE) and other medical board exams.

**CommonsenseQA** is a multiple choice benchmark testing commonsense reasoning, with challenging associations extracted from concept net to find many target concepts for a single source concept.

**OpenbookQA** is a multiple choice dataset requiring multi-step reasoning over common and commonsense knowledge requiring deeper understanding of a diverse set of topics.

**MMLU** is a massive benchmark covering 57 subjects from a diverse set of areas including STEM, humanities, and social sciences. This benchmark tests both more rudiementary and more advanced sets of knowledge for these topics covering great breadth and depth.

## A.2 MODEL ACCURACIES

Following are the accuracies for each of the models on the 12 datasets. Answers are elicited using the prompt format specified in A.3. As expected the GPT-4 model has the highest accuracies on all 12 datasets, followed by Claude v1.3. Llama-2-Chat and Base have comparable accuracy to GPT-3.5. Text-davinci-003 has the lowest accuracies.

| Model | TQA | MedQA | CSQA | OBQA | Law | Ethics | Physics |
|---|---|---|---|---|---|---|---|
| Text-davinci | 0.472 | 0.504 | 0.712 | 0.772 | 0.532 | 0.590 | 0.579 |
| Llama-2 | 0.524 | 0.564 | 0.664 | 0.808 | 0.572 | 0.590 | 0.583 |
| Llama-2 Chat | 0.480 | 0.512 | 0.684 | 0.728 | 0.528 | 0.600 | 0.528 |
| GPT-3.5 | 0.572 | 0.536 | 0.676 | 0.776 | 0.504 | 0.670 | 0.523 |
| Claude | 0.596 | 0.640 | 0.736 | 0.832 | 0.604 | 0.760 | 0.638 |
| GPT-4 | **0.836** | **0.736** | **0.768** | **0.940** | **0.664** | **0.760** | **0.813** |

| | Econ | Algebra | Chem | Security | Policy | Avg |
|---|---|---|---|---|---|---|
| Text-davinci | 0.412 | 0.300 | 0.440 | 0.690 | 0.850 | 0.571 |
| Llama-2 | 0.386 | 0.240 | 0.460 | 0.750 | 0.910 | 0.588 |
| Llama-2 | 0.333 | 0.310 | 0.420 | 0.670 | 0.850 | 0.554 |
| GPT-3.5 | 0.404 | 0.320 | 0.460 | 0.730 | 0.800 | 0.581 |
| Claude | 0.579 | 0.330 | 0.500 | 0.760 | 0.880 | 0.655 |
| GPT-4 | **0.596** | **0.480** | **0.520** | **0.800** | **0.910** | **0.735** |

Table 4: **Model Accuracies** Accuracies of all 6 models on all 12 tasks. GPT-4 is the highest performing model for all tasks.

## A.3 LINGUISTIC CONFIDENCE PROMPTS

We study 24 different prompt formats to elicit linguistic confidences. These include prompts eliciting numerical confidences (score from 0-1, 0-10, 0-100), probabilities from 0-100%, linguistic categorization of confidences into varying numbers of confidence buckets ('not sure', 'sure', 'very sure'), eliciting chain-of-thought explanations of confidence in addition to confidence scores, and varying the confidence instructions. We measure the AUC performance of each prompt across the 12 datasets for all models, and select the prompt which reduces the sum of drops in AUC from the best prompt for each individual model. This prompt which we use for all of our experiments is described below (we chose this because it led to highest accuracy and near 0 extraction errors):

Answer the following question to the best of your ability, and provide a score between 0 and 1 to indicate the confidence you have in your answer. Confidence scores closer to 0 indicate you have less confidence in your answer, while scores closer to 1 indicate you have more confidence in your answer. You must answer the question with one of the valid choices. You must provide only a single answer.

Question: This is a question
(A) first answer
(B) second answer
(C) third answer
(D) fourth answer
(E) fifth answer
Answer: (D)
Confidence: 0.4

Question: This is another question
(A) first answer
(B) second answer
(C) third answer
(D) fourth answer
(E) fifth answer
Answer: (A)
Confidence: 0.7

Question: {question}
Answer:

Following are a few of the several alternative prompts we tried:

Answer the following question to the best of your ability. Indicate your confidence in your answer by saying 'I am very sure.', 'I am sure.', 'I am not sure.'. Please provide the letter of your answer choice and your confidence assessment in the following format:
Answer: (your answer)
Confidence assessment: (your confidence score)

Question: {question}

Answer the following question to the best of your ability, and provide a probability between 0 and 100%, indicating the likelihood that your answer is correct. Please provide the letter of your answer choice and your probability estimate in the following format:
Answer: (your answer)
Probability of correctness: (your probability estimate)

Question: {question}

## A.4 COMPARISONS WITH OTHER CERTAINTY SCORERS

While Lin et al. (2023) also study uncertainty quantification for black-box models, their work primarily focuses on NLG tasks – quantifying uncertainty by sampling generations and computing similarity scores. We instead study confidence estimation for discriminative tasks like MC question answering. Xiao et al. (2022) study reducing calibration error through choice of LLM, uncertainty quantifier, and fine-tuning loss. We study much larger LLMs, and do not study confidence estimation in a fine-tuning setting. Their work also suggests using temperature scaling for confidence estimates. We experiment with Platt scaling, but do not report results since it improves ECE but does not change AUC or AUROC, since scaling confidences does not affect their relative ordering.

## A.5  LLAMA-2-70B CHAT RESULTS

We also experiment with the chat version of the Llama-2-70B model, evaluating both the AUC and AUROC for its linguistic confidences and model probabilities. We find that the Base version of the model slightly outperforms the Chat version in both linguistic confidences and model probabilities.

| Metric | Confidence Type | TQA | MedQA | CSQA | OBQA | Law | Ethics | Physics |
|--------|-----------------|-----|-------|------|------|-----|--------|---------|
| AUC | Llama-2 Chat Linguistic | 0.574 | 0.520 | 0.685 | 0.742 | 0.529 | 0.630 | 0.561 |
|  | Llama-2 Chat Prob | **0.696** | **0.604** | **0.840** | **0.869** | **0.674** | **0.823** | **0.721** |
| AUROC | Llama-2 Chat Linguistic | 0.679 | 0.517 | 0.506 | 0.535 | 0.501 | 0.562 | 0.568 |
|  | Llama-2 Chat Prob | **0.750** | **0.609** | **0.764** | **0.776** | **0.710** | **0.821** | **0.721** |

| Metric | Confidence Type | Econ | Algebra | Chem | Security | Policy | Avg |
|--------|-----------------|------|---------|------|----------|--------|-----|
| AUC | Llama-2 Chat Linguistic | 0.357 | 0.304 | 0.442 | 0.698 | 0.846 | 0.574 |
|  | Llama-2 Chat Prob | **0.438** | **0.348** | **0.632** | **0.850** | **0.963** | **0.705** |
| AUROC | Llama-2 Chat Linguistic | 0.553 | 0.485 | 0.546 | 0.560 | 0.479 | 0.541 |
|  | Llama-2 Chat Prob | **0.634** | **0.495** | **0.721** | **0.811** | **0.858** | **0.722** |

Table 5: **AUC and AUROC Metrics for Llama-2-70b-chat linguistic confidences** and model probabilities.

## A.6  ECE RESULTS

We compute the expected calibration error metric (ECE) by dynamically binning examples based on their confidence scores into 10 bins with approximately equal numbers of examples. For each bin, compute the calibration error, which is the absolute difference between the mean predicted probability and the observed accuracy. This quantifies how well the predicted probabilities match the true probability of the positive class within each bin. We then calculate the weighted average of the calibration errors across all bins, where the weights are the proportion of examples in each bin relative to the total number of examples. A lower ECE indicates better calibration. If the predicted probabilities are well-calibrated, the ECE should be close to zero. Following are the ECEs for the linguistic confidence scores of each model and the ECEs of model probabilities for models which provide them.

### A.6.1  GPT-4 ALL CONFIDENCE METHODS

Following are the ECEs for each confidence method for the GPT-4 model. For 11 out of 12 tasks we find that one of our proposed methods leads to the lowest ECE value.

| | TQA | MedQA | CSQA | OBQA | Law | Ethics | Physics |
|--|-----|-------|------|------|-----|--------|---------|
| Ling. Conf. | 0.104 | 0.118 | 0.118 | 0.038 | 0.187 | 0.114 | 0.109 |
| SC Ling. Conf. | 0.126 | 0.163 | 0.120 | 0.036 | 0.246 | 0.204 | 0.120 |
| Surrogate[†] | 0.395 | 0.212 | 0.297 | 0.370 | 0.156 | 0.205 | 0.317 |
| Tiebreak[†] | 0.114 | 0.134 | 0.126 | **0.032** | 0.194 | 0.114 | 0.118 |
| Mixture[†] | 0.096 | **0.075** | **0.061** | 0.159 | **0.064** | **0.111** | **0.088** |
| SC Mixture[†] | **0.085** | 0.120 | 0.108 | 0.029 | 0.216 | 0.186 | 0.098 |

| Confidence Type | Econ | Algebra | Chem | Security | Policy | Avg |
|-----------------|------|---------|------|----------|--------|-----|
| Ling. Conf. | 0.270 | 0.420 | 0.313 | 0.118 | **0.053** | 0.164 |
| SC Ling. Conf. | 0.323 | 0.379 | 0.331 | 0.136 | 0.063 | 0.187 |
| Surrogate[†] | 0.129 | **0.162** | **0.187** | 0.210 | 0.264 | 0.242 |
| Tiebreak[†] | 0.270 | 0.419 | 0.332 | 0.158 | 0.068 | 0.173 |
| Mixture[†] | **0.126** | 0.224 | 0.229 | **0.108** | 0.138 | **0.123** |
| SC Mixture[†] | 0.287 | 0.358 | 0.286 | 0.129 | 0.068 | 0.164 |

Table 6: **ECE Values All Confidence Methods for GPT-4**

### A.6.2 Comparing Linguistic Confidences and Model Probabilities

| Confidence Type | TQA | Medqa | CSQA | OBQA | Law | Ethics | Physics |
|---|---|---|---|---|---|---|---|
| Text-davinci Linguistic | **0.422** | **0.425** | **0.161** | **0.127** | **0.380** | **0.300** | **0.299** |
| Text-davinci Prob | 0.461 | 0.454 | 0.235 | 0.191 | 0.388 | 0.338 | 0.338 |
| Llama-2 Linguistic | 0.365 | 0.248 | 0.201 | **0.073** | 0.224 | 0.259 | 0.267 |
| Llama-2 Prob | **0.099** | **0.084** | **0.176** | 0.235 | **0.115** | **0.145** | **0.094** |
| Llama-2 Chat Linguistic | 0.357 | 0.391 | 0.125 | 0.101 | 0.350 | **0.194** | 0.337 |
| Llama-2 Chat Prob | **0.284** | **0.228** | **0.124** | **0.092** | **0.264** | 0.213 | **0.210** |
| GPT-3.5 Linguistic | 0.350 | 0.380 | 0.192 | 0.091 | 0.388 | 0.176 | 0.363 |
| Claude Linguistic | 0.187 | **0.086** | **0.042** | **0.033** | **0.098** | **0.052** | 0.162 |
| GPT-4 Linguistic | **0.104** | 0.118 | 0.118 | 0.038 | 0.187 | 0.114 | **0.109** |

| Confidence Type | Econ | Algebra | Chem | Security | Policy | Avg |
|---|---|---|---|---|---|---|
| Text-davinci Linguistic | **0.482** | 0.625 | 0.475 | **0.213** | **0.038** | **0.329** |
| Text-davinci Prob | 0.478 | **0.576** | **0.385** | 0.263 | 0.112 | 0.352 |
| Llama-2 Linguistic | 0.453 | 0.561 | 0.435 | **0.079** | **0.093** | 0.272 |
| Llama-2 Prob | **0.205** | **0.091** | **0.100** | 0.172 | 0.264 | **0.148** |
| Llama-2 Chat Linguistic | 0.505 | 0.480 | 0.480 | **0.165** | **0.055** | 0.295 |
| Llama-2 Chat Prob | **0.403** | **0.361** | **0.272** | 0.187 | 0.073 | **0.226** |
| GPT-3.5 Linguistic | 0.515 | 0.560 | 0.432 | 0.173 | **0.094** | 0.309 |
| Claude Linguistic | **0.132** | **0.319** | **0.175** | **0.058** | 0.162 | **0.126** |
| GPT-4 Linguistic | 0.270 | 0.420 | 0.313 | 0.118 | 0.053 | 0.164 |

Table 7: **ECE Values Linguistic Confidences vs Model Probabilities**

### A.7 Surrogate Confidences vs Sampled Linguistic Confidences

**Surrogate confidences are cheaper and more reliable than sampling.** While black-box models do support sampling, sampling to get linguistic confidences is actually far more expensive than our proposed surrogate model method. With the best performing sampling method (SC Hybrid Ling. Conf., Table 3) eliciting linguistic confidences (Xiong et al., 2023) from GPT-4 requires sampling *five* times from GPT-4, while our surrogate method requires sampling only *once* from GPT-4 for an answer and *once* from a much smaller and cheaper model like Llama-2-70B for a confidence — so our surrogate method significantly reduces computational cost and complexity over sampling.

**Sampling for confidences only works well with additional post-processing.** The best performing sampling method (SC Hybrid Ling. Conf., Table 3) requires further updating and post-processing steps on top of the sampled confidence scores, adding additional complexity and reducing interpretability. It is also unclear if sampled confidences only perform well in conjunction with Xiong et al. (2023)'s update rule or if the performance can generalize to other forms of post-processing.

To study how well sampling for confidences performs without post-processings, we sample confidences from GPT-4, following Xiong et al. (2023)'s procedure (sampling 5 times at $T = 0.7$ and applying self-consistency), and applying no additional post-processing steps to the sampled confidences (SC Vanilla Ling. Conf., Table 3). We find that across 12 datasets, vanilla confidence sampling *significantly underperforms* our surrogate model method for GPT-4 — resulting in an average AUC of only 77.3% (compared to 81.7% average AUC for our surrogate method, and 83% for our mixture method) and an average AUROC of only 59.3% (compared to 65.4% for our surrogate method and 69.6% for our best mixture method) (Table 3).

**Surrogate confidences can be combined with sampled confidences for further gains.** Surrogate confidences are complementary to the sampled, post-processed confidences from Xiong et al. (2023). Interestingly, we are able to derive further improvements in confidence estimation by composing the surrogate probabilities and sampled, post-processed linguistic confidences (SC

mixture) — average AUC of 84.5% and average AUROC of 74%, with up to 6% improvement in AUC and 7% improvement in AUROC for individual tasks (Table 3).

### A.8 OPTIMIZING ALPHA FOR MIXTURE OF MODELS

Following are the optimal values of $\alpha$ for the mixture of models method (Algorithm 1) using the best surrogate model (Llama-2-70b Base).

|  | GPT-3.5-Turbo | Claude-v1.3 | GPT-4 (Mixture) | GPT-4 (SC Mixture) |
|---|---|---|---|---|
| $\alpha$ | 0.6 | 0.6 | 0.4 | 0.3 |

Table 8: **Optimal Alphas for Mixture of Models**

We select a single $\alpha$ for each model to be used across tasks, by sweeping over values of $\alpha$ from 0 to 1 at increments of 0.1 and optimizing for the value which leads to the highest AUC per model, averaged over the 12 tasks. Results reported in Table 2 and Table 3 leverage these values of $\alpha$.

### A.9 GOOD SURROGATES MAKE SIMILAR MISTAKES AS MAIN MODELS

We generate embeddings for questions that GPT-4 answers incorrectly, questions that a strong surrogate, Llama-2-70B, answers incorrectly, and finally questions that a weaker surrogate, Llama-2-13B, answers incorrectly. We use OpenAI's embedding API to generate these embeddings using the text-embedding-ada-002 model. Any model of reasonable quality could be used to produce these embeddings. We then use PCA to represent the embeddings in a 2D space for visualization. Finally, we plot the embeddings of questions answered incorrectly by these models from two representative datasets, TruthfulQA and MMLU - College Chemistry, and study the semantic similarity of mistakes made as approximated by the 2D spatial similarity of embeddings of their incorrectly answered questions (Figure 4). See Figure 4 for further details on how to interpret the plots. We find that there is greater semantic similarity in the mistakes made by GPT-4 and Llama-2-70B, than those made by GPT-4 and Llama-2-13B — suggesting that GPT-4 and Llama-2-70B may find similar questions to be difficult, allowing Llama-2-70B's confidence scores to better transfer to GPT-4.

### A.10 COMBINING MULTIPLE SURROGATES

The key contribution of the mixture of models technique (Section 5) was demonstrating that confidence signals from different models are *composeable*. We expect the optimal composition of surrogate models to vary depending on the task and the main model used to produce the answers.

**Experiments.** To investigate the utility of combining multiple surrogate models, we compute confidence estimates for GPT-4 using a linear regression model to learn a weighted combination of confidence scores from multiple surrogates. The weights are *not* constrained to be positive or to sum to one. We conduct these experiments on two representative datasets, MedQA and CommonsenseQA, and explore combinations of the following surrogate confidences: Llama-2-70B Base probabilities, Llama-2-70B Chat probabilities, GPT-4 linguistic confidences, GPT-4 hybrid self-consistency linguistic confidences (Section 5), Claude-v1.3 linguistic confidences, and GPT-3.5-Turbo linguistic confidences. The regression model is trained on 500 examples, and we evaluate on 250 held-out examples for each dataset.

**Results.** For MedQA, we find that composing surrogate confidences from multiple models (probabilities from Llama-2-70B Base and Chat, linguistic confidences from Claude-v1.3, and hybrid SC linguistic confidences from GPT-4) with GPT-4's linguistic confidence scores leads to a 1.8% improvement in AUC and a 4% improvement in AUROC over just composing Llama 2 70B's probabilities with GPT-4's linguistic confidences (Table 3). However, for CommonsenseQA, we find that composing confidences from multiple surrogate models harms confidence estimation compared to just composing Llama-2-70B's surrogate probabilities with GPT-4's confidences (Table 3). This suggests that the benefits of composing multiple surrogate models are task dependent — there may be more value to combining multiple surrogates for tasks where the differences in confidence signals from models are substantive without encoding noise.

### A.11 EXTENDED SURROGATE MODEL RESULTS

We present detailed results on the effect of varying surrogate confidence models on the AUC and AUROC of the corresponding main models.

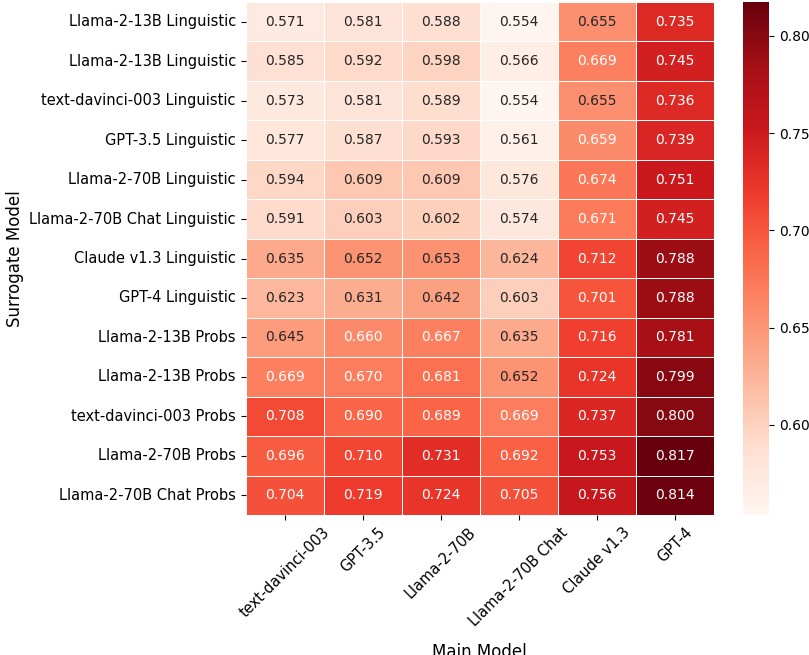

Figure 5: **AUCs For All Surrogate Models** We compute the AUC metric for each model considering surrogate confidences from both model probabilities and linguistic confidence scores from all other models. We find that all models benefit from using surrogate model probabilities over their own linguistic confidences.

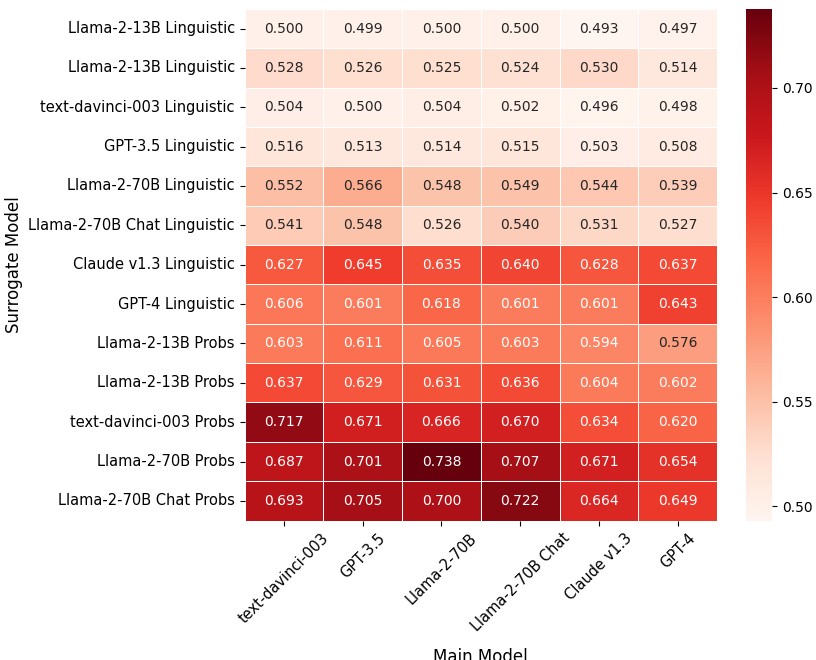

Figure 6: **AUROCs For All Surrogate Models** We also compute the AUROC metric for each model considering surrogate confidences from both model probabilities and linguistic confidence scores from all other models. In general, we find that using surrogate model probabilities instead of a model's own linguistic confidences improves AUROC values.

## A.12 ANALYSIS

We compute correlations and covariances between the answer correctness (set of binary scores indicating a model answered a question correctly or not) for every pair of main model and potential surrogate model. We find that in general if a surrogate model $S$ has a high degree of correlation in answer correctness with a main model $M$, then $S$ is likely to be a good surrogate for $M$. Similar trends also hold for covariances.

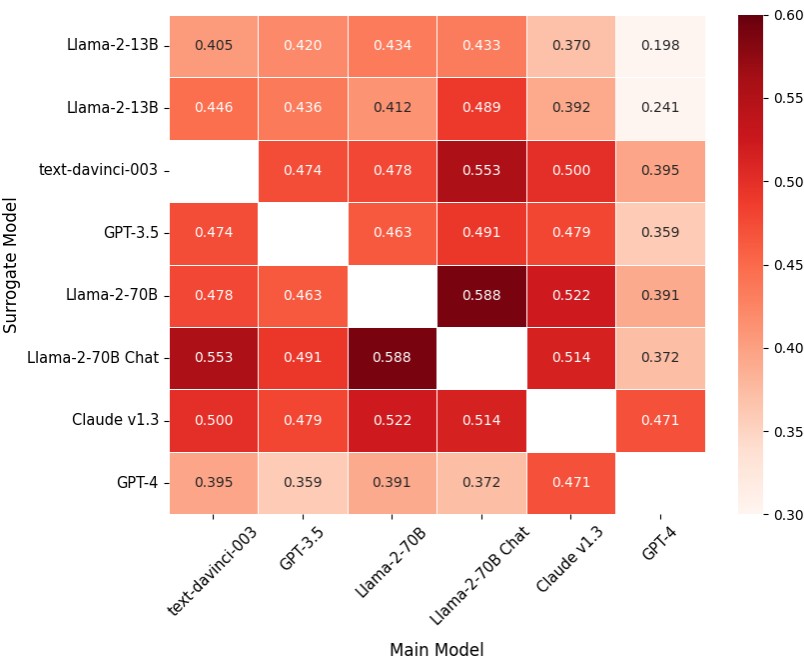

Figure 7: **Correlations For All Main and Surrogate Models**

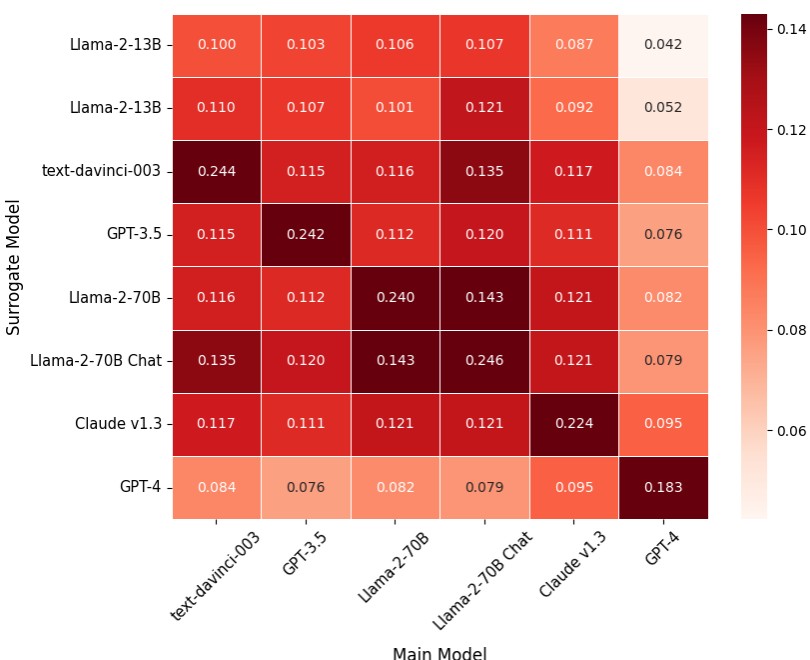

Figure 8: **Covariances For All Main and Surrogate Models**

## A.13 CALIBRATION OF MIXTURE OF MODELS

We also plot the distribution of confidence scores in equally spaced buckets from 0 to 1 and the corresponding accuracies of those buckets in reliability diagrams to study the change in calibration between linguistic confidences and our best mixture of models methods and find that our methods better calibrate the confidence scores.

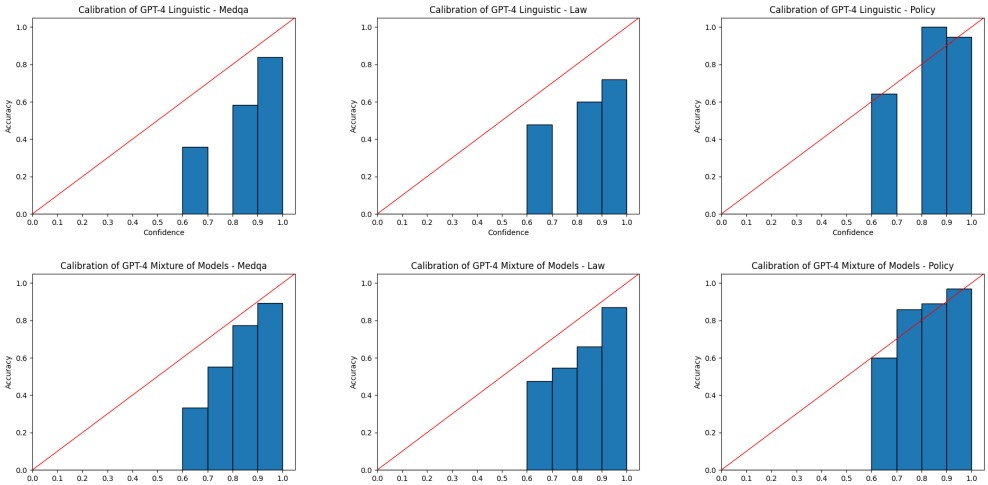

Figure 9: **Calibration of GPT-4 with Linguistic Confidence and Mixture of Models** In the first row we see the calibration of GPT-4 on MedQA, Professional Law, and US Foreign Policy when using linguistic confidences. In the second row, we see GPT-4's calibration using our mixture of models confidence method. A perfectly calibrated model would have all bars aligned with the red line. We can see that calibration improves demonstrably, when using mixture of models.

## A.14 CORRELATION BETWEEN LINGUISTIC CONFIDENCES AND MODEL PROBABILITIES

| Dataset | Llama-2-70B Base | Llama-2-70B Chat |
|---------|------------------|------------------|
| TQA | 0.357 | 0.293 |
| MedQA | 0.176 | 0.174 |
| CSQA | 0.308 | 0.166 |
| OBQA | 0.218 | 0.251 |
| Law | 0.041 | 0.120 |
| Ethics | 0.407 | 0.210 |
| Physics | 0.310 | 0.227 |
| Econ | 0.266 | 0.137 |
| Algebra | 0.134 | -0.222 |
| Chem | 0.173 | -0.063 |
| Security | 0.477 | 0.325 |
| Policy | 0.355 | 0.254 |
| **Avg** | **0.269** | 0.156 |

Table 9: **Correlation Between Linguistic Confidences and Model Probabilities**

We measure the Pearson correlation coefficient between linguistic confidences and model probabilities for models that provide access to internal probabilities and report the correlation coefficient for each model and task in Table 9. We do not report results for text-davinci-003 because for many tasks it outputs the same linguistic confidence score for each example, resulting in an undefined correlation coefficient. Both the Base and Chat Llama-2-70B models have a slight positive correlation in their linguistic confidences and model probabilities. We observe that the correlation between

linguistic confidences and model probabilities is stronger for Llama-2-70B Base, than for Llama-2-70B Chat — average Pearson correlation coefficient of 0.269 vs 0.156. However, the correlation is not very strong for either model indicating that linguistic confidences do not fully capture the confidence signal of model probabilities.

### A.15 CONFIDENCE ESTIMATION AND SELF-CRITIQUING

Several studies have demonstrated that LLMs can generate feedback and self-improve (Madaan et al., 2023; Chang et al., 2023). However, other research has indicated that these models may not be proficient at self-evaluation, as they can sometimes hallucinate or generate incorrect feedback (Huang et al., 2023). The definition of self-evaluation is largely dependent on the task at hand. For instance, in creative, open-ended generation tasks such as story writing, there may not always be a clear standard of correctness, so the goal of self-evaluation or feedback generation may be to iteratively improve some other aspect of the generation. However, in cases where model generations are more factual — such as using an LLM to generate a math proof and allowing it to critique and iterate on each of its reasoning steps — calibrated confidence estimation becomes more important for self-evaluation. If a model could produce good quality confidence estimates for its self-critiques, these could guide the model in deciding when to trust and act on its own feedback, thereby enhancing its performance and reliability.

### A.16 RANDOMIZED AUC RESULTS

**AUC with Randomized or Deterministic Classifiers.** To plot the accuracy-coverage curve we compute $A(c)$, the selective accuracy at coverage $c$ across different values of $c$. $A(c)$ is the accuracy if the model only makes a prediction on the $c$ proportion of data with highest confidence scores. for different values of $c$. When making a prediction on $c$ proportion of data, for each example $x$ we use a binary classifier on the confidence score $C(x)$ decide if we are making a prediction for $x$ (1 if making a prediction and 0 if abstaining from prediction). Such a classifier can either be deterministic or randomized.

*Deterministic Classifiers.* A deterministic classifier $f$ returns identical outputs for identical inputs — resulting in consistent treatment of examples with the same confidence score (either predict on all or abstain on all). Using a deterministic classifier to select $c$ portion of examples to predict on means we find the highest confidence threshold $t$ such that $P(C(x) \geq t) \geq c$ — $t$ is the highest confidence threshold where the proportion of examples with confidence greater than or equal to $t$ is greater than or equal to the required coverage $c$. With a deterministic classifier, we predict on $P(C(x) \geq t)$ proportion of examples, which may be greater than the required coverage $c$.

$$f(C(x)) \in \{0, 1\} \tag{A.1}$$

*Randomized Classifiers.* A randomized classifier $h$ can return different outputs for the same input. Since models can output the same linguistic confidences for multiple examples, a randomized classifier can allow us to achieve exactly a coverage of $c$ by making predictions on some examples with a given confidence, and abstaining on other examples with the same confidence. To enable $h$ to break ties and make different predictions for examples with the same confidence score, we add a small amount of Gaussian noise to each confidence score $\mathcal{N}(0, \epsilon), \epsilon \to 0$ to enforce a confidence-based ordering of examples.

$$h(C(x) + \mathcal{N}(0, \epsilon)) \in \{0, 1\} \tag{A.2}$$

*Deterministic vs Randomized AUC Example.* Suppose a model assigns half of the examples a confidence of 1 and gets them all right, and the other half of examples a confidence of 0.5 and gets 50% of them right. What is the selective accuracy at coverage 75%? A deterministic classifier would select 0.5 as $t$ and predict on all examples with $C(x) \geq t$, which in this case is all of the examples (notably leading to a coverage of 100% instead of 75%). This would lead to an accuracy of 75%. A randomized classifier would predict on all examples of confidence 1, but to meet the 75% coverage threshold, it would predict on half of the examples which have confidence 0.5 —

| | Confidence Type | TQA | MedQA | CSQA | OBQA | Law | Ethics | Physics |
|---|---|---|---|---|---|---|---|---|
| **AUC** | Text-davinci Linguistic | 0.523 | 0.504 | 0.718 | 0.775 | 0.532 | 0.590 | 0.579 |
| | Text-davinci Prob | **0.607** | **0.656** | **0.861** | **0.929** | **0.714** | **0.783** | **0.697** |
| | Llama 2 Linguistic | 0.600 | 0.616 | 0.693 | 0.802 | 0.605 | 0.707 | 0.638 |
| | Llama 2 Prob | **0.711** | **0.735** | **0.804** | **0.923** | **0.749** | **0.834** | **0.763** |
| | GPT-3.5 Linguistic | 0.620 | 0.536 | 0.693 | 0.776 | 0.508 | 0.674 | 0.526 |
| | Claude-v1.3 Linguistic | 0.741 | 0.718 | **0.807** | 0.879 | 0.669 | **0.894** | 0.736 |
| | GPT-4 Linguistic | **0.889** | **0.841** | 0.802 | **0.960** | **0.732** | 0.869 | **0.819** |
| **AUROC** | Text-davinci Linguistic | 0.525 | 0.500 | 0.503 | 0.509 | 0.500 | 0.500 | 0.500 |
| | Text-davinci Prob | **0.718** | **0.696** | **0.806** | **0.840** | **0.715** | **0.758** | **0.637** |
| | Llama 2 Linguistic | 0.618 | 0.541 | 0.555 | 0.484 | 0.517 | 0.602 | 0.593 |
| | Llama 2 Prob | **0.745** | **0.722** | **0.731** | **0.777** | **0.733** | **0.868** | **0.732** |
| | GPT-3.5 Linguistic | 0.535 | 0.500 | 0.526 | 0.518 | 0.508 | 0.509 | 0.504 |
| | Claude-v1.3 Linguistic | **0.701** | 0.586 | **0.639** | 0.647 | **0.586** | **0.760** | **0.652** |
| | GPT-4 Linguistic | 0.665 | **0.716** | 0.551 | **0.656** | 0.591 | 0.720 | 0.522 |

| | Confidence Type | Econ | Algebra | Chem | Security | Policy | Avg |
|---|---|---|---|---|---|---|---|
| **AUC** | Text-davinci Linguistic | 0.412 | 0.300 | 0.440 | 0.690 | 0.856 | 0.577 |
| | Text-davinci Prob | **0.431** | **0.338** | **0.644** | **0.891** | **0.939** | **0.707** |
| | Llama 2 Linguistic | 0.415 | 0.189 | 0.474 | 0.817 | 0.930 | 0.624 |
| | Llama 2 Prob | **0.498** | **0.263** | **0.647** | **0.866** | **0.981** | **0.731** |
| | GPT-3.5 Linguistic | 0.430 | 0.319 | 0.465 | 0.724 | 0.806 | 0.590 |
| | Claude-v1.3 Linguistic | 0.640 | 0.333 | 0.653 | 0.812 | 0.934 | 0.735 |
| | GPT-4 Linguistic | **0.643** | **0.551** | **0.683** | **0.903** | **0.965** | **0.805** |
| **AUROC** | Text-davinci Linguistic | 0.500 | 0.500 | 0.500 | 0.500 | 0.506 | 0.504 |
| | Text-davinci Prob | **0.549** | **0.532** | **0.695** | **0.858** | **0.795** | **0.717** |
| | Llama 2 Linguistic | 0.533 | 0.424 | 0.520 | 0.613 | 0.576 | 0.548 |
| | Llama 2 Prob | **0.622** | **0.546** | **0.732** | **0.775** | **0.871** | **0.738** |
| | GPT-3.5 Linguistic | 0.518 | 0.522 | 0.505 | 0.519 | 0.519 | 0.515 |
| | Claude-v1.3 Linguistic | **0.573** | 0.543 | 0.708 | 0.687 | 0.645 | 0.644 |
| | GPT-4 Linguistic | 0.551 | **0.599** | **0.721** | **0.750** | **0.753** | **0.650** |

Table 10: **AUC and AUROC - Linguistic Confidences vs Model Probabilities** We compare the AUC and AUROC values for linguistic confidences and model probabilities in weaker models (text-davinci-003 and Llama 2 70B), and find that model probabilities consistently outperform linguistic confidences. For closed source models (which don't provide model probabilities), we see that Claude-v1.3 and GPT-4 provide the best linguistic confidences in both AUC and AUROC.

selecting the top half after adding random noise. This would lead to an accuracy of approximately 83%.

Our main results are presented computing AUC with a deterministic classifier, in accordance with several works in the selective classification space (El-Yaniv & Wiener, 2010; Liang et al., 2022). AUC computed with a randomized or deterministic classifier would be equivalent if confidence estimates for all examples were distinct. But since models can output the same linguistic confidences for multiple examples, and randomized AUC computation can improve the AUC in these cases, we also present results with a randomized classifier. Using a randomized classifier leads to slight improvements for both linguistic confidences and our mixture of models method (since it incorporates linguistic confidences), but does not change any of our qualitative results.

| | | Text-davinci | GPT-3.5 | Llama 2 | Claude-v1.3 | GPT-4 |
|---|---|---|---|---|---|---|
| **AUC** | Ling. Conf. | 0.577 | 0.590 | 0.624 | 0.735 | 0.805 |
| | Surrogate† | 0.707 | 0.719 | **0.731** | 0.763 | 0.821 |
| | Tiebreak† | **0.711** | 0.719 | 0.715 | 0.764 | 0.830 |
| | Mixture of Models† | **0.711** | **0.722** | **0.731** | **0.772** | **0.834** |
| **AUROC** | Ling. Conf. | 0.504 | 0.514 | 0.548 | 0.637 | 0.646 |
| | Surrogate† | 0.717 | 0.708 | **0.738** | 0.671 | 0.657 |
| | Tiebreak† | **0.718** | 0.708 | 0.699 | 0.683 | 0.692 |
| | Mixture of Models† | **0.718** | **0.709** | 0.737 | **0.687** | **0.699** |

Table 11: **AUC and AUROC of Surrogate and Mixture of Model Methods.** We compare the performance of our proposed methods† with the baseline linguistic confidence method (gray). For both AUC and AUROC, our proposed methods outperform linguistic confidences on all models. Mixture of models improves the AUC of GPT-4 by 3% and AUROC by 5%.

| | Method | TQA | MedQA | CSQA | OBQA | Law | Ethics | Physics |
|---|---|---|---|---|---|---|---|---|
| **AUC** | Ling. Conf. | 0.889 | 0.841 | 0.802 | 0.960 | 0.732 | 0.869 | 0.819 |
| | SC Ling. Conf. | 0.903 | **0.887** | 0.841 | 0.978 | 0.729 | **0.902** | 0.846 |
| | Surrogate† | 0.866 | 0.844 | 0.849 | 0.965 | 0.762 | 0.849 | **0.891** |
| | Tiebreak† | 0.902 | 0.871 | 0.833 | 0.967 | 0.768 | 0.889 | 0.861 |
| | Mixture† | 0.895 | 0.864 | 0.849 | 0.969 | **0.780** | 0.882 | 0.886 |
| | SC Mixture† | **0.921** | 0.873 | **0.877** | **0.979** | 0.757 | 0.894 | 0.881 |
| **AUROC** | Ling. Conf. | 0.665 | 0.716 | 0.551 | 0.656 | 0.591 | 0.720 | 0.522 |
| | SC Ling. Conf. | 0.698 | **0.767** | 0.625 | 0.833 | 0.619 | **0.817** | 0.592 |
| | Surrogate† | 0.543 | 0.666 | 0.656 | 0.683 | 0.619 | 0.617 | 0.648 |
| | Tiebreak† | 0.671 | 0.750 | 0.611 | 0.716 | 0.628 | 0.740 | 0.589 |
| | Mixture† | 0.642 | 0.731 | 0.646 | 0.731 | 0.655 | 0.711 | 0.648 |
| | SC Mixture† | **0.702** | 0.747 | **0.679** | **0.838** | **0.655** | 0.783 | **0.663** |

| | Method | Econ | Algebra | Chem | Security | Policy | **Avg** |
|---|---|---|---|---|---|---|---|
| **AUC** | Ling. Conf. | 0.643 | 0.551 | 0.683 | 0.903 | 0.965 | 0.805 |
| | SC Ling. Conf. | 0.663 | 0.584 | 0.726 | 0.915 | 0.965 | 0.828 |
| | Surrogate† | **0.667** | 0.572 | 0.724 | 0.888 | 0.971 | 0.821 |
| | Tiebreak† | 0.654 | 0.580 | 0.746 | 0.910 | 0.974 | 0.830 |
| | Mixture† | 0.664 | 0.581 | 0.749 | 0.908 | **0.976** | 0.834 |
| | SC Mixture† | 0.662 | **0.645** | **0.763** | **0.926** | 0.973 | **0.846** |
| **AUROC** | Ling. Conf. | 0.551 | 0.599 | 0.721 | 0.750 | 0.753 | 0.650 |
| | SC Ling. Conf. | 0.622 | 0.682 | 0.818 | 0.798 | 0.755 | 0.719 |
| | Surrogate† | 0.578 | 0.621 | 0.706 | 0.779 | 0.764 | 0.657 |
| | Tiebreak† | 0.569 | 0.648 | 0.760 | 0.815 | 0.805 | 0.692 |
| | Mixture† | 0.578 | 0.648 | 0.759 | 0.814 | **0.822** | 0.699 |
| | SC Mixture† | **0.595** | **0.763** | **0.819** | **0.839** | 0.810 | **0.741** |

Table 12: **AUC and AUROC of All Confidence Methods for GPT-4.** Our proposed surrogate model method outperforms linguistic confidences on 9/12 datasets on AUC. Mixing surrogate probabilities and linguistic confidences outperforms vanilla linguistic confidences on AUC for all 12 datasets. The mixture of surrogate probabilities also outperforms hybrid self-consistency confidences, the best method in Xiong et al. (2023), on average (AUC 83.4% vs 82.8%. Mixing surrogate probabilities with self-consistency linguistic confidences leads to the best confidence estimates overall, outperforming all methods with an average 84.6% AUC and 74.1% AUROC, which is a gain of 4.1% and 9.1% respectively over vanilla linguistic confidences.

