# OpenReview forum: "Llamas Know What GPTs Don't Show: Surrogate Models for Selective Classification"
_ICLR.cc/2024/Conference — ICLR 2024 Conference Withdrawn Submission_

### Official Review · Reviewer_LeSy · 2023-10-29

**Soundness:** 3 good
**Presentation:** 3 good
**Contribution:** 3 good
**Rating:** 5
**Confidence:** 4

**Summary:**

This paper studies the role of confidence in improving the LLM performances in QA tasks. Some LLMs do not output confidences, so a new method to elicit the confidence is necessary.

This paper first considers “linguistic confidence”: prompt the model to output a notion of confidence. However, for the models that provides accesses to probabilities, the performances from the linguistic confidence are worse than the performances from the probability confidence.

This paper proposes using surrogate models (i.e., some models where we do have access to their probabilities) to estimate the confidence. On 10 out of 12 datasets, the method has a higher AUC than the “linguistic confidence” method.

**Strengths:**

- The proposed confidence estimation approach is elegant. It’s simple, and it works well.
- There are extensive experiments showing that the proposed approach (and its variants) work on a wide range of problems.
- There are also extensive ablation studies showing the different combinations of the surrogate models. (Some studies are not covered though — please refer to my comment below.)

**Weaknesses:**

- The proposed algorithm seems to have limitations. The proposed algorithm 1 still requires the main model to output linguistic confidences (unless alpha=1), which is a confidence score less as good as the probability scores.
- The evaluation can be more rigorous. I have been looking for the evaluations for the validity of the linguistic confidence. Specifically, how well do they correlate to the probabilities directly outputted by the models? The evaluation scores presented in this paper focused on the utility of these confidence scores though.
- The value of a crucial hyperparameter is not reported. Alpha, the scaling factor between the two confidence scores, seems very important for the overall AUC / AUROC performances. The actual values for the optimal settings, or the approaches to reach the optimal values, are not reported.
    - A related note, the heading of the second paragraph in 5.1 (”Epsilon is all you need”) seems to indicate that very small alpha values are sufficient, which is obviously not the case, considering that “Tiebreak” and “Surrogate” settings have quite different results. In general, a claim like “XYZ is all you need” usually leave me with a (perhaps wrong here) impression that the paper is a social media post rather than a scientific paper.
- An intuitive extension of the algorithm could have been explored. Since using one surrogate works well, does combining two surrogate models work?

**Questions:**

What do the “+” symbols at the end of many methods in tables 2 and 3 mean?

---

> ### Author Response · Authors · 2023-11-23
> **Official Comment by Authors**
>
> We thank the reviewer for the constructive feedback and questions, and for expressing that our experiments and ablation studies are “extensive”, and that our approach is “elegant” and “works well”. Following we respond to each of the reviewer’s comments — discussing why linguistic confidences are useful for mixture of models,  adding experiments combining multiple surrogate models, highlighting ECE measurements for linguistic confidences, adding analysis of correlation between linguistic confidences and model probabilities, specifying the exact values of alpha and how to arrive at them, and providing other clarifications.
> Please let us know if you have any additional questions or comments. Thank you!
>
> > “algorithm 1 still requires the main model to output linguistic confidences (unless alpha=1), which is a confidence score less as good as the probability scores.”
>
> **Linguistic confidences lower bound the confidence, surrogate probabilities improve the bound with fine-grained signal:** The reviewer is correct in stating that the algorithm 1 (mixture of models) approach leverages linguistic confidence scores, which do not *independently* provide confidence estimates as good as probabilities from surrogate models. However these confidences are still useful when *combined* with surrogate probabilities. The variant of algorithm 1, which we term 'tiebreak' (adding only a small proportion of surrogate probabilities to linguistic confidences), provides intuitions on why linguistic confidences are valuable when composed with surrogate probabilities. Linguistic confidences even for a strong model like GPT-4 are not expressive — GPT-4 provides a confidence score of 0.9 for 50% of its examples across 12 datasets (Section 6). However, when linguistic confidences are combined with surrogate probabilities, **linguistic confidences have an anchoring effect, essentially serving as a 'lower bound' estimate of the confidence of each example, as assessed by the main model** (which is answering the question). **Adding surrogate probabilities to these confidences then provides controlled modulation of the confidence score**, adding a more fine-grained signal which then allows a relative ordering of confidence estimates for examples which previously had the same linguistic confidences.
>
> > “Does combining two surrogate models work?”
>
> We appreciate this suggested extension to our mixture of models method. The key contribution of the mixture technique was demonstrating that confidence signals from different models are composeable – so our initial focus had been on solidifying this finding with a single surrogate model.
>
> **Additional experiments with multiple surrogate models:** We ran additional experiments on two representative datasets MedQA and CommonsenseQA — for each task we compute confidence estimates for GPT-4 using a linear regression model to learn a weighted combination of confidences from multiple surrogate models: Llama-2-70B probabilities, Llama-2-70B chat probabilities, GPT-4 hybrid self-consistency linguistic confidences (Section 5, Table 3), Claude-v1.3 linguistic confidences, GPT-3.5-Turbo linguistic confidences.
>
> For MedQA, we find that **composing surrogate confidences from multiple models** (all models except GPT-3.5-Turbo) with GPT-4’s confidence scores leads to a **1.8% improvement in AUC and a 4% improvement in AUROC** over just composing Llama-2-70B’s probabilities with GPT-4.
>
> However, for CommonsenseQA, we find composing confidences from multiple surrogate models harms confidence estimation compared to just composing Llama-2-70B’s surrogate probabilities with GPT-4’s confidences.
>
> This implies that the **benefits of composing multiple surrogate models are task dependent**— there may be more value to combining multiple surrogates for tasks where the differences in confidence signals from models are substantive but do not encode noise. We include more details on the experiments and our results in Appendix A.10.

---

> ### Author Response · Authors · 2023-11-23
> **Official Comment by Authors**
>
> > “evaluations for the validity of the linguistic confidence.”
>
> Though there can be different interpretations of the notion of validity of confidence scores, based on the comment we presume that the reviewer may be referring to an evaluation of the ECE (calibration error) of the linguistic confidence scores. We report ECE measurements for both linguistic confidences and model probabilities in Appendix A.6. Based on the ECE scores, we see that on average, for **Llama-2 linguistic confidences are less well calibrated than the model’s probabilities, but for text-davinci-003 linguistic confidences are slightly better calibrated than the model’s probabilities.** Specific ECE values for all confidence scorers can be found in Table 6 of Appendix A.6.
>
> We would like to emphasize that we focus our analysis on the AUROC and AUC metrics (which measure how useful a confidence is in distinguishing between correct and incorrect examples) due to the following shortcoming of the ECE metric - “Intuitively, calibration requires that if we output a 0.6 confidence on 100 examples, then we should get 0.6 · 100 = 60 of them correct. For a classifier with accuracy A, one (degenerate) way to have perfect calibration (best possible ECE) is to output confidence = A for every example.” (footnotes of Section 2) In other words, **while calibration can be improved by bringing confidence scores close to the task accuracy, this doesn’t improve confidence utility.**
>
> > “how well do they [linguistic confidences] correlate to the probabilities directly outputted by the models?”
>
> We measure the **Pearson correlation coefficient between linguistic confidences and model probabilities** for models that provide access to internal probabilities and report the correlation coefficient for each model and task in Table 9. We do not report results for text-davinci-003 because for many tasks it outputs the same linguistic confidence score for each example, resulting in an undefined correlation coefficient.
> Both the Base and Chat Llama-2-70B models have a slight positive correlation in their linguistic confidences and model probabilities. We observe that the **correlation between linguistic confidences and model probabilities is stronger for Llama-2-70B Base**, than for Llama-2-70B Chat --- average Pearson correlation coefficient of 0.269 vs 0.156. However, **the correlation is not very strong for either model, indicating that linguistic confidences do not fully capture the confidence signal of model probabilities.** We have added this analysis to Appendix A.14.

---

> ### Author Response · Authors · 2023-11-23
> **Official Comment by Authors**
>
> > Exact values of $\alpha$ and process to arrive at them
>
> We apologize for not detailing the exact values of $\alpha$ (weight on the surrogate probabilities in the mixture algorithm - Algorithm 1). We placed less importance on the exact value, since it could differ depending on the selected surrogate model. The optimal values of alpha using our best surrogate model (Llama-2-70B) are:
>
> |                  | GPT-3.5-Turbo  | Claude-v1.3  |  GPT-4 (Mixture)  |  GPT-4 (SC Mixture)  |
> | ----------- | -----------------  | --------------- | ------------------- | ------------------------ |
> | $\alpha$   |        0.6       | 0.6 | 0.4 | 0.3 |
>
> We select the alpha hyperparameter by **sweeping over values from 0 to 1 at increments of 0.1 and optimizing for the alpha which leads to the highest AUC value per model**, averaged over 12 datasets. We have added these details to Appendix A.8 of the paper.
>
> > Confusion in wording that “very small alpha values are sufficient”
>
> We appreciate the reviewer’s note and have updated the header of the subsection to be 'Why Does Tiebreaking Work Well' in Analysis (Section 6) accordingly. Although the absolute AUC and AUROC metrics for the 'tiebreak' and 'mixture of model' methods are different, we mention that small alpha values are sufficient given that **for GPT-4 tiebreaking is able to achieve 90% of the AUC gains and 86% of the AUROC gains that the mixture method produces** over linguistic confidences. Furthermore, tiebreaking is highlighted in this paper to provide an **intuitive understanding of why combining linguistic confidences and surrogate probabilities is valuable** — linguistic confidence provides a lower bound anchor of the confidence for each example and even incorporating a small fraction of surrogate probabilities can help modulate the linguistic confidence score and hone in on a more fine-grained confidence value.
>
> > “What do the “+” symbols at the end of many methods in tables 2 and 3 mean?”
>
> We apologize for the confusion – the dagger (+) symbol after the method names in Tables 2 and 3 is used to indicate the methods we propose. Conversely, the grayed out rows in the tables are the baseline results we compare against.

---

### Official Review · Reviewer_RfxS · 2023-10-31

**Soundness:** 2 fair
**Presentation:** 2 fair
**Contribution:** 2 fair
**Rating:** 5
**Confidence:** 3

**Summary:**

This paper leverages the open-source LLM known as "llama 2" to assess the uncertainty or confidence of outputs from the black-box LLM model, GPT-4. The authors demonstrate that by using the llama 2 confidence scorer, one can achieve a higher AUC. Moreover, the paper introduces a novel mixture function designed to combine outputs from multiple confidence-scorer models, ultimately resulting in an optimized scorer.

**Strengths:**

- Assessing the uncertainty of black-box language models represents a significant and intriguing research direction.
- Leveraging the probability metrics from an open-sourced language model is an intuitive approach.
- The authors provide comprehensive AUC results from a variety of confidence-scorer models and policy models. These findings will be valuable for future researchers when choosing a confidence scorer.

**Weaknesses:**

- Soundness: The methodological soundness of this study appears somewhat lacking. There's a noticeable lack of a baseline comparison in the work.
  - While the paper primarily focuses on uncertainty or confidence, it doesn't compare with established certainty scorers. It would be beneficial to discuss relevant works such as [1] and [2] and incorporate them in the experimental section.
  - The study also touches on the critiquability of LLMs and LLM evaluation. Including references [3], [4], and [5] in the related work and experiments would provide more depth and context to the discussions.

- Novelty and Contribution:
  - The approach of using surrogate models to interpret black-box models isn't novel.
  - The introduced mixture function, essentially a simple linear combination, raises questions regarding its uniqueness. A clearer differentiation from existing methods might strengthen this section.

- Clarity and Writing Quality:
  - The manuscript could benefit from further editing for clarity and structure. For detailed feedback, refer to the 'Question' section.

[1] Uncertainty Quantification with Pre-trained Language Models:A Large-Scale Empirical Analysis

[2] Generating with Confidence: Uncertainty Quantification for Black-box Large Language Models

[3] Self-Refine: Iterative Refinement with Self-Feedback

[4] CRITIC: Large Language Models Can Self-Correct with Tool-Interactive Critiquing

[5] A Survey on Evaluation of Large Language Models

**Questions:**

In Table 1, for every row, is the policy model identical to the scorer model, effectively making it a self-scorer? As an instance, does "Text-davinci Prob" employ "Text-davinci" as both its policy and scorer model?

Regarding the statement "embeddings of questions that GPT-4 gets incorrect" – can you provide clarity on how these embeddings are derived or obtained?

---

> ### Author Response · Authors · 2023-11-23
> **Official Comment by Authors**
>
> We thank the reviewer for the thoughtful feedback and questions, expressing that our work studies a “significant and intriguing research direction”, that our techniques are “intuitive”, our results are “comprehensive”, and that our “findings will be valuable for future researchers”. Below we have tried to address each of the reviewer’s comments — comparing our work with more types of certainty scorers, further discussing the novelty and uniqueness of our approach, contrasting with the LLM self-evaluation space, and providing a few clarifications.
> Please let us know if you have any additional questions or comments. Thank you!
>
> > “paper primarily focuses on uncertainty or confidence, it doesn't compare with established certainty scorers”
>
> We would like to highlight that we do **include comparisons against Xiong et al. 2023 [3]**, who study sampling for linguistic confidences and aggregating them using self-consistency. These baselines are included in Table 3 (SC Vanilla Ling. Conf. — sampled confidence scores, and SC Hybrid Ling. Conf. — sampled confidence scores with additional post-processing). **Xiong et al.’s methods are *significantly* more expensive** than ours, since they involve prompting GPT-4 multiple times and add further complexity by requiring post-processing steps. **Our surrogate and mixture methods are cheaper** in involving a single query to GPT-4 for the answer, and sampling a smaller surrogate model like Llama-2-70B just once for a confidence estimate. Our methods also **outperform Xiong et. al.’s work** (Section 5). For a more detailed comparison of our work against these sampling baselines see our Related Work and Appendix A.7.
>
> That said, we thank you for sharing references [1] and [2] on uncertainty quantification! We have carefully reviewed these references, and added a reference to Appendix A.4 in our Related Work for a more detailed comparison with these works. We summarize the comparison as follows:
>
> While [2] also studies uncertainty quantification for black-box models, this paper primarily focuses on NLG tasks by sampling generations and computing similarity scores. Our work instead focuses on **uncertainty quantification for discriminative tasks like multiple-choice question answering** – the discriminative application of [2] would be akin to the sampling baselines from [3] which we compare against in Table 3.
> [1] studies different parts of an LLM based prediction pipeline designed to reduce calibration error – a) choice and size of LLM, b) choice of uncertainty quantifier, and c) finally choice of fine-tuning loss. Comparisons with a) are not applicable to our work, since **all of our experiments are done on LLMs larger and more performant than [1]’s recommended model (ELECTRA)**. Comparisons with c) are also not applicable, since **our setting involves confidence elicitation from models without further fine-tuning or adaptation.** [1] suggests using temperature scaling for b) uncertainty quantification. Temperature scaling is a single-parameter variant of platt-scaling — **we experiment with platt-scaling, but do not report results since it improves ECE but does not change AUC or AUROC** (since scaling confidences doesn’t affect their relative ordering).
>
> **References:**
>
> [1] Uncertainty Quantification with Pre-trained Language Models: A Large-Scale Empirical Analysis. Y. Xiao et al. 2022.
>
> [2] Generating with Confidence: Uncertainty Quantification for Black-box Large Language Models. Z. Lin et al. 2023.
>
> [3] Can LLMs Express Their Uncertainty? An Empirical Evaluation of Confidence Elicitation in LLMs. M. Xiong et al. 2023.

---

> ### Author Response · Authors · 2023-11-23
> **Official Comment by Authors**
>
> > “approach of using surrogate models to interpret black-box models isn't novel”
>
> We have reviewed the references suggested by the reviewer, however none of these seem to discuss the use of surrogate models to interpret black-box models. [1] focuses on optimizing different parts of an NLP pipeline for better calibration, [2] studies uncertainty quantification for black-box models on NLG tasks through sampling generations and estimating uncertainty through dissimilarity in generations, and [3]-[5] focus on LLM self-evaluation. We welcome references to other works that use surrogate models for black-box model interpretation. In the meantime, **we share some more details on why we feel our contributions are scientifically interesting and novel**, as well as future directions we expect our work to inspire.
>
> **Transferability of Confidences Between Models of Different Families and Sizes:** A surrogate model (like Llama-2-70B) can generate **high-quality confidence estimates for a different main model (like GPT-4), despite being weaker and coming from a different model family**. This outcome is unexpected and novel, given that surrogate confidences can transfer to a main model without requiring any additional fine-tuning or other adaptations based on the main model. We study different sizes of surrogates (Llama-2-70B and Llama-2-13B) and demonstrate that similarity in size (to the main model) and accuracy of the surrogate may contribute to the efficacy of confidence transfer (Section 4).
>
> **Future Directions:** Different aspects of LLMs may lead to better transferability of confidence scores – including the **training data, their fine-tuning regime (the effect of instruction-tuning), their sizes, architectures, or model families**. We provide initial results on the effect of model size (Llama-2-70B vs Llama-2-13B) and family (transfer of text-davinci-003 to GPT-4) in Section 4, and hope that our findings can inspire future work along the other mentioned directions.
>
> > “Mixture function raises questions about uniqueness”
>
> While the mixture function is simple, the main technical contribution of our mixture of models method is not the exact function implementation, but instead the finding that **confidence signals from different models are complementary and can be composed.** This result is surprising and significant because it is not apparent why combining confidences from *separate* models would lead to improved confidence estimates for either model. Given the transferability of confidences demonstrated by the surrogate model method, we might instead expect that different models express the *same* notion of uncertainty through their confidences. However if this were the case we would not expect better confidence estimates by combining scores from different models. The fact that we do see better confidence estimation through both the surrogate and mixture of model methods suggests that confidence signals from models do contain related information (allowing surrogate to transfer), but not identical information. Moreover, the **differences in the signals, far from being irrelevant noise, are complementary and advantageous for the main model.**
>
> **References:**
>
> [1] Uncertainty Quantification with Pre-trained Language Models: A Large-Scale Empirical Analysis. Y. Xiao et al. 2022.
>
> [2] Generating with Confidence: Uncertainty Quantification for Black-box Large Language Models. Z. Lin et al. 2023.
>
> [3] Self-Refine: Iterative Refinement with Self-Feedback. A. Madaan et al. 2023.
>
> [4] CRITIC: Large Language Models Can Self-Correct with Tool-Interactive Critiquing. Z. Gou et al. 2023.
>
> [5] A Survey on Evaluation of Large Language Models. Y. Chang et al. 2023.

---

> ### Author Response · Authors · 2023-11-23
> **Official Comment by Authors**
>
> > “study also touches on the critiquability of LLMs and LLM evaluation”, the reviewer suggests contrasting our work with others in the LLM evaluation domain
>
> Thank you for highlighting the connection of our work to recent works in the LLM self-evaluation and self-critique domains. We have updated our Related Work section and Appendix A.15 to discuss the suggested references and contextualize our work with regards to the self-evaluation space. We provide a summary below:
>
> LLMs have shown progress in self-critiquing and improving using their own feedback or other tools [1], [2]. This self-evaluation can have different goals depending on the task at hand — for example a story writing task may require more feedback about creativity over factuality. However, **producing good quality confidence estimates can be useful for tasks where feedback is used to self-correct a model’s generations**. A key challenge in self-evaluation is model generated feedback can include factual inaccuracies, logical errors, or other hallucinations and therefore it can be difficult to use this feedback to correct the original generation [3]. Other works have focused on sampling many generations and directly used model probabilities as a signal of correctness [4], [5]. **Better confidence estimates can allow the model to rank and select feedback more likely to be correct, or directly rank its generations based on likelihood of correctness.**
>
> > “In Table 1, for every row, is the policy model identical to the scorer model, effectively making it a self-scorer?”
>
> We apologize for the confusion in interpreting the Table 1 results. In the terminology used by the reviewer, **for the results presented in Table 1, each policy model is indeed a self-scorer.** For models which do provide access to internal probabilities (text-davinci-003, Llama 2), we provide results scoring the model’s confidence with both its prompted linguistic confidences (e.g. Text-davinci Linguistic for text-davinci-003) and scoring the model’s confidence with the probability it places on its outputted answer choice (e.g. Text-davinci Prob). For black-box models which currently do not provide access to internal probabilities (gpt-3.5-turbo-0613, Claude-v1.3, GPT-4), we score the model’s confidence only using prompted linguistic confidences (e.g. Claude Linguistic). We hope this clarifies the Table 1 results and we have modified Table 1’s caption to better describe this.
>
> > “Regarding the statement "embeddings of questions that GPT-4 gets incorrect" – can you provide clarity on how these embeddings are derived or obtained?”
>
> In Figure 4, we generate embeddings for questions that GPT-4 answers incorrectly, questions that a strong surrogate, Llama-2-70B, answers incorrectly, and finally questions that a weaker surrogate, Llama-2-13B, answers incorrectly. We use **OpenAI’s embedding API to generate these embeddings using the text-embedding-ada-002 model**, although any model of reasonable quality could be used to produce these embeddings. We then use **PCA to represent the embeddings in a 2D space** for visualization. Finally, we plot the embeddings of questions answered incorrectly by these models on **two representative datasets, TruthfulQA and MMLU - College Chemistry**, and study the semantic similarity of mistakes these models make as approximated by the 2D spatial similarity of embeddings of their incorrectly answered questions Figure 4 provides further details on how to interpret these plots. We find that there is greater *semantic similarity* in the mistakes made by GPT-4 and Llama-2-70B, than those made by GPT-4 and Llama-2-13B — suggesting that **GPT-4 and Llama-2-70B may find similar questions to be difficult, allowing Llama-2-70B’s confidence scores to better transfer to GPT-4**. We have added these details to Appendix A.9.
>
> **References**
>
> [1] Self-Refine: Iterative Refinement with Self-Feedback. A. Madaan et al. 2023.
>
> [2] CRITIC: Large Language Models Can Self-Correct with Tool-Interactive Critiquing. Z. Gou et al. 2023.
>
> [3] Large Language Models Cannot Self-Correct Reasoning Yet. J Huang et al. 2023.
>
> [4] Self-Evaluation Guided Beam Search for Reasoning. Y. Xie et al. 2023.
>
> [5] Language Models (Mostly) Know What They Know. S. Kadavath et al. 2022.

---

### Official Review · Reviewer_mEr2 · 2023-11-02

**Soundness:** 2 fair
**Presentation:** 2 fair
**Contribution:** 2 fair
**Rating:** 3
**Confidence:** 3

**Summary:**

This paper proposes to use open-source models such as LLaMa as a proxy for finding confidence estimates for models that do not provide probabilities, such as GPT or Claude.

**Strengths:**

The paper has fairly extensive experimentation over a large number of tasks.

**Weaknesses:**

I feel that this method introduces additional complexity in a place where it is not clearly needed, and because of this I am skeptical of whether this method will see wide adoption should the paper be accepted to ICLR.

Specifically, I am not convinced of the underlying premise of the paper, that you cannot get probabilities out of closed models. Specifically, it is well known that sampling can be used to approximate probabilities (see the "Pattern Recognition and Machine Learning" textbook for example), and all closed models that I know of support sampling. Xiong et al. empirically demonstrated that this is a quite effective way of getting probability estimates out of models, and this is much easier than additionally running a separate proxy model to get probability estimates. The mixture of surrogate probabilities method indeed marginally beats the best method of Xiong et al. (by 0.4% AUC for example), but this doesn't seem to warrant the additional complexity.

**Questions:**

None in particular, although I would be open to arguments about why this method may be preferable over other simpler alternatives.

---

> ### Author Response · Authors · 2023-11-23
> **Official Comment by Authors**
>
> We thank the reviewer for the thoughtful feedback and questions, and expressing that our work has an “extensive experimentation over a large number of tasks”. Below we have tried to address the reviewer’s comments by providing a detailed discussion on why our surrogate and mixture methods are preferable to sampling for confidences from black-box models. We welcome any additional questions or comments. Thank you!
>
> To summarize:
> - sampling for confidences only works with post-processing steps which add further complexity
> - surrogate confidences are *cheaper* than sampling and perform *better*
> - our method can be combined with sampling for *further gains*
> - the success of surrogate confidences highlights a deeper *transferability* between models, which we believe is of scientific interest to the community
> This discussion has also been added to Appendix A.7 of our work.
>
> > “sampling can be used to approximate probabilities…this is much easier than additionally running a separate proxy model to get probability estimates”, “[surrogate method] doesn't seem to warrant the additional complexity”
>
> **Surrogate confidences are *cheaper* and perform *better* than sampling:**
> The reviewer is correct in mentioning that closed models do support sampling. However sampling to get linguistic confidences is actually far more expensive than our proposed surrogate model method. Xiong et al., 2023’s [1] best performing method requires sampling five times from an expensive model like GPT-4, while our surrogate method requires sampling only once from GPT-4 for an answer and once from a much smaller and cheaper model like Llama-2-70B for a confidence — so our surrogate method in fact *significantly reduces* computational cost and complexity. Our method is both far cheaper for users and produces better confidence estimates than sampling — we believe that these are important and sufficient reasons for our method to see widespread adoption.
>
> **Sampling only works well with *additional* post-processing of confidences:**
> Xiong et al.’s best results (SC Hybrid Ling. Conf., Table 3) require further updating and post-processing steps on top of the sampled confidence scores, adding further complexity to their method and making it difficult to interpret the sampled scores. The motivation behind their particular choice of update rule is also unclear, making it difficult to understand how it was derived or if sampled confidences can only perform well in conjunction with their specific post-processing steps.
>
> **To study how well sampling for confidences performs on its own, we add a new baseline (SC Vanilla Ling. Conf., Table 3)** by sampling confidences from GPT-4, following Xiong et al’s procedure (sampling 5 times at T=0.7 and applying self-consistency), and applying no additional post-processing to the sampled confidences. We find that **across 12 datasets, vanilla sampling *significantly underperforms* our surrogate model method for GPT-4** — resulting in an average AUC of only 77.3% (compared to 81.7% average AUC for our surrogate method, and 84.5% for our best mixture method) and an average AUROC of only 59.3% (compared to 65.4% for our surrogate method and 74% for our best mixture method). Detailed metrics for each dataset with this new baseline are included in Table 3 in Section 5.
>
> **Our method can be combined with sampled confidences for further gains:**
> The reviewer cites a 0.4% improvement in GPT-4’s average AUC of our mixture method over the sampling + post-processing baseline (SC Hybrid Ling. Conf., Table 3).  However, as we demonstrate in section 5, surrogate confidences are complementary to the sampled, post-processed confidences from Xiong et al. (SC Mixture, Table 3) Interesting because of the complementary nature of surrogate probabilities and linguistic confidences, we are able to derive further improvements in confidence estimation by composing the two — **average AUC of 84.5% and average AUROC of 74%, with up to 6% improvements in AUC and 7% improvements in AUROC for individual tasks.**
>
> **References:**
>
> [1] Can LLMs Express Their Uncertainty? An Empirical Evaluation of Confidence Elicitation in LLMs. M. Xiong et al. 2023.

---

> ### Author Response · Authors · 2023-11-23
> **Official Comment by Authors**
>
> **Success of surrogate confidences highlights a *transferability phenomenon* between models:**
>
> Additionally, we feel that it is crucial to highlight that our findings are also important and interesting from a scientific perspective — they shine light on an aspect of *transferability* that exists between language models. It is unexpected and surprising that a transfer of confidence estimates between *different* language models (different sizes, different model families) should hold. We are among the first papers to demonstrate such transfer and provide intuitions and analysis for why this transfer may occur (Section 6). We are optimistic that our results will inspire future work on understanding the transferability between language models along several other axes such as the effect of fine-tuning regimes, model training data, and model architecture. As explored in our work, this transferability between models can allow the use of white-box models to understand different aspects of black-box models.

---

### Official Review · Reviewer_78gw · 2023-11-03

**Soundness:** 2 fair
**Presentation:** 3 good
**Contribution:** 2 fair
**Rating:** 3
**Confidence:** 4

**Summary:**

The paper is focused on confidence elicitation for models that do not provide confidence probabilities their answers. Such models include GPT-3.5., GPT-4 and Claude. Linguistic confidences are obtained by zero-shot prompting the models to assign confidence scores to their answers. The linguistic confidences are evaluated in a selective classification setting (where the goal is to have confidence scores that are calibrated with the correctness of the answers). Two metrics are used for evaluation: AUC (area under the coverage-accuracy curve) and AUROC (area under the receiver operator curve). Experimental results using 12 standard question answering datasets show that the linguistic confidences are not much better than random guesses. Furthermore, they are worse than model probabilities from surrogate models such as Llama-2 variants. The best results are obtained when linguistic confidences are mixed with surrogate model probabilities.

**Strengths:**

The paper includes an extensive number of experiments over 12 standard question answering datasets and the results are consistent over the 12 datasets.

It is interesting to know that surrogate model probabilities are a better indicator of confidence than the linguistic confidences. It's also interesting to see that combining surrogate model probabilities and linguistic confidences improves the results.

**Weaknesses:**

The paper mainly consists of a large set of well-conducted experiments, but lacks the depth.

While the results are interesting, they are actually not very surprising. The mixture of models approach is very straightforward.

The discussion of the results is not very insightful. Given the focus of the paper, it would be interesting to better understand the reasons the models are not good at eliciting good linguistic confidence scores. While the authors claim that error calibration is not the focus of the paper, it would be interesting to know how uncalibrated surrogate models perform by comparison with calibrated surrogate models.

**Questions:**

N/A

---

> ### Author Response · Authors · 2023-11-23
> **Official Comment by Authors**
>
> We thank the reviewer for the constructive feedback, and for expressing that our work “includes an extensive number of experiments”, our “results are consistent over the 12 datasets”, and that our results are interesting. Below we have tried to address all of the reviewer’s comments – discussion of why our results are surprising, what the effect of surrogate model calibration is, and why eliciting good linguistic confidences is difficult. Please let us know if you have any additional questions or comments. Thank you!
>
> > “While the results are interesting, they are actually not very surprising.”
>
> We believe it is surprising that a *surrogate* model is able to produce well-calibrated confidence estimates for a *different main* model (based on AUC, AUROC, and ECE). This is an unexpected result because the surrogate model is able to make good confidence predictions for the main model — **without requiring any fine-tuning or other adaptation** based on the main model. The **surrogate model producing these confidences (e.g., Llama-2-70B for GPT-4) is a *weaker* model than GPT-4 and comes from a different model family**, so it is even more surprising that it is nonetheless able to produce well-calibrated confidences for a stronger model. This indicates a deeper *transferability* that exists between language models, which can allow us to use white-box models, like Llama-2, to better understand black-box models, like GPT-4.
>
> > “The mixture of models approach is very straightforward.”
>
> It is critical to note that the appeal of the mixture method is that **confidence signals from different models are complementary and composeable**. This result is surprising and noteworthy, regardless of the composition function we use, because it is not clear why composing the confidences of different models should be useful for either model.
>
> In fact, given that surrogate confidences transfer well, one may instead expect the confidence scores from *different* models to encode the exact *same* information. But if this were the case then composing the confidence signals of different models would not produce *better* confidence estimates, similar to how composing a model’s confidence with its own confidence would not yield additional benefits. Interestingly, the success of both the surrogate method and the mixture method implies that signals from a main model and its surrogate do encode related information (allowing surrogate confidences to transfer), but not identical information — and furthermore that, instead of being inconsequential noise, the differences in the signals are complementary and beneficial to the main model.
>
> > “would be interesting to know how uncalibrated surrogate models perform by comparison with calibrated surrogate models”
>
> Calibration (ECE) of a surrogate model can be greatly improved using recalibration techniques like platt-scaling. However, platt-scaling does not affect selective classification performance since it does not modify the relative order of confidence scores (if confidence A > B, then platt-scaling retains this order). So the **calibration (based on ECE) of a surrogate will not factor into how useful it is in improving the selective accuracy of a different model**. However, the *selective classification performance* of a surrogate may impact its utility as a surrogate — for example, Llama-2-70B has a better AUC than text-davinci-003 (73.1% vs 70.8%, Table 1) and is a better surrogate for most models (based on AUC and AUROC, Figure 2 and Appendix A.11).

---

> ### Author Response · Authors · 2023-11-23
> **Official Comment by Authors**
>
> > “Would be interesting to better understand the reasons the models are not good at eliciting good linguistic confidence scores.”
>
> Below we describe some reasons for why eliciting good linguistic confidences from LLMs is difficult. A summary of these points is also highlighted in our Analysis section (Section 6).
>
> **Challenges in Producing Diverse Generations:** When prompted zero-shot, models tend to generate confidences from a small discrete set which hinders good AUC performance, since assigning the same confidence to many examples makes it difficult to use confidences to distinguish between correct and incorrect examples. Even the strongest model we work with, GPT-4, produces a confidence of 0.9 for *50% of examples* across 12 datasets. Through experiments with a comprehensive set of 24 prompts (Appendix A.2), we empirically demonstrate that eliciting a diverse set of confidences from the model is difficult regardless of prompt format. This relates to the standard challenge in natural language generation tasks of models producing repetitive, non-diverse generations — as models tend to repeat the same words and phrases, we see that they also repeat the same confidences.
>
> **Limitations of Training Corpora:** There are imbalances in the frequencies of numerical values in training corpora. Zhou et al., 2023 [1] conduct analysis of the Pile and find that the use of 'nice' percentages like 50%, 95%, and 100% frequently occurs in the dataset. This may be due to discussion of confidence intervals (95% confidence interval) or exaggerated expressions of confidence ('I’m 100% sure!') (Zhou et al). So it is possible that training corpora may not contain good representations of low confidences causing models to primarily output higher confidence scores.
>
> **Difficulty in Understanding 'Confidence’ Terminology:** Eliciting good zero-shot confidences requires the models to 1) linguistically understand what it means to be confident about an answer, 2) be able to internally assess their confidence, and 3) represent this confidence linguistically. These requirements make linguistic confidence elicitation an inherently challenging task. To mitigate the difficulty of 1), we experiment with prompts describing confidence through different words (uncertainty, probability of correctness, confidence etc.), with different levels of detail in the task instruction, and allowing for different representations of confidence (scores from 0-1, % probability of correctness, chain of thought confidence description) but find that elicitation of high-quality confidences remains a difficult task.
>
> **Effects of RLHF:** We experiment with the Llama-2-70B Base model as well as the Llama-2-70B Chat model (Section 4, Appendix A.5). For most tasks, we find that the Base model is better at producing linguistic confidences, while the chat model produces more overconfident estimates. This finding may suggest that RLHF tuning pushes models to be more compliant to users — displaying higher levels of certainty could be more appealing or reassuring to users.
>
> **References**
>
> [1] Navigating the Grey Area: How Expressions of Uncertainty and Overconfidence Affect Language Models. K. Zhou et al. 2023.

---

### Author Response · Authors · 2023-11-23
**Overall Response**

We thank the reviewers for their time and helpful comments and feedback. The reviewers agree that
we study a “significant and intriguing research direction” (RfxS) with “extensive experimentation over a large number of tasks” (mEr2), that our “results are interesting” (78gw) and “findings will be valuable for future researchers” (RfxS), and that our approach is “elegant” and “works well” (LeSy).

(i.) Contribution and novelty: We introduce the surrogate model method of confidence estimation for state-of-the-art models which currently do not provide access to model probabilities or internal representations. We find that **surprisingly a *weaker* model from a *different model* family (Llama-2-70B) is able to provide good confidence estimates for a stronger model like GPT-4** – without requiring any fine-tuning or further adaptation. We also propose the mixture of models method which shows that **confidence signals from different models are surprisingly *complementary* and *composeable*** — leading to state-of-the-art confidence estimates for GPT-4 and Claude-v1.3.

(ii.) Comparison with related work: Other works have elicited confidence estimates by sampling linguistic confidences and applying self-consistency. These methods however are quite expensive in needing to sample large black-box models like GPT-4 *multiple* times, and do not work without additional post-processing steps, which add further complexity.
**Our method is simpler, cheaper, and more effective than sampling for confidences** — we sample *once* from GPT-4 for an answer and *once* from a much smaller model (like Llama-2) for a confidence, and are able to achieve better confidence estimates. Additionally, our methods can be combined with self-consistency-based confidences for even better confidence estimation. (Section 5, Appendix A.7)

(iii.) Future directions: Our work also **highlights a deeper *transferability* between language models** that can allow for the use of white-box models like Llama-2 to better understand black-box models like GPT-4. We are optimistic that our findings will inspire future work on better understanding how model similarity in other dimensions impacts transferability — different fine-tuning regimes, model training data, and model architecture.

Thank you to all of the reviewers for their constructive suggestions! We have incorporated their feedback as follows, and believe this makes our contributions even stronger:
- Additional baseline on sampling for confidences without post-processing in Section 5, Table 3.
- Detailed comparison of our methods with sampling for confidences in Appendix A.7.
- Additional experiments combining multiple surrogate models in Appendix A.10.
- Additional analysis on the correlation between linguistic confidences and model probabilities in Appendix A.14.
- Comparison with more certainty scorers in Appendix A.4 and contrasting with the LLM self-evaluation domain in Related work, Appendix A.15.
- Details on the exact values of alpha for the mixture of models method and how to arrive at them in Appendix A.8.
- Details on generating embeddings of incorrectly answered questions in Appendix A.9.

More details can be found in our response to each reviewer. We also present results with a variant of the AUC metric in Appendix A.16 for completeness – notably this does not change our findings.